# Experimental evidence that viscous shear zones generate periodic pore sheets

James Gilgannon[1], Marius Waldvogel[1], Thomas Poulet[2], Florian Fusseis[3], Alfons Berger[1], Auke Barnhoorn[4], and Marco Herwegh[1]

[1]Institute of Geological Sciences, University of Bern, 3012 Bern, Switzerland
[2]CSIRO Mineral Resources, Kensington, WA 6151, Australia
[3]School of Geosciences, The University of Edinburgh, Edinburgh EH9 3JW, UK
[4]Department of Geoscience and Engineering, Delft University of Technology, Delft, The Netherlands

**Correspondence:** James Gilgannon (james.gilgannon@geo.unibe.ch)

**Abstract.**

In experiments designed to understand deep shear zones, we show that periodic porous sheets emerge spontaneously during viscous creep and that they facilitate mass transfer. These findings challenge conventional expectations of how viscosity in solid rocks operates and provide quantitative data in favour of an alternative paradigm, that of the dynamic granular fluid pump model. On this basis, we argue that our results warrant a reappraisal of the community's perception of how viscous deformation in rocks proceeds with time and suggest that the general model for deep shear zones should be updated to include creep cavitation. Through our discussion we highlight how the integration of creep cavitation, and its Generalised Thermodynamic paradigm, would be consequential for a range of important solid Earth topics that involve viscosity in Earth materials, like slow earthquakes, the flow of glacial ice and the tectonics of exoplanets.

## 1 Introduction

Our existing models for mantle convection, the advance of glaciers and even the dynamics of the seismic cycle all include, and rely on, the concept that solids can be viscous and flow with time. In this sense, the fluid mechanical concept of viscosity is a cornerstone of Geoscience and our view of a dynamic Earth is built around it. In rocks, a record of this viscosity is found in mylonitic shear zones, the largest of which are the deep boundaries of tectonic plates that can reach into the asthenospheric upper mantle (Vauchez et al., 2012). Consequentially, mylonites represent important interfaces in the lithosphere that crosscut different geochemical, geophysical and hydrological domains. This role places them at the centre of discussions on slow earthquakes and the hydrochemical exchange of deep and shallow reservoirs (e.g. Beach, 1976; Fusseis et al., 2009; Bürgmann, 2018). In this context, it is critical to have a robust and complete model of deep shear zones and the viscous rocks in them.

The accepted conceptual model for lithospheric shear zones supposes that there is a mechanical stratification with depth from an upper frictional to lower viscous domain (Sibson, 1977; Schmid and Handy, 1991; Handy et al., 2007). In this model, viscous creep is a continuous slow background deformation and, at certain conditions, is punctuated by fracturing. It is this

fracturing, which can have physical (e.g. Beall et al., 2019) or chemical (e.g. Alevizos et al., 2014) driving forces, that creates seismicity and mass transport pathways through the deep Earth (Sibson, 1994). Two core assumptions of this conceptual model are that creep in polycrystalline aggregates mainly contributes to distorting the deforming mass (e.g. Poirier, 1985; Hobbs and Ord, 2015) and the large confining pressures of the viscous domain reduce porosity and permeability with compaction (Edmond and Paterson, 1972; Xiao et al., 2006). In contrast, there is a newer paradigm that argues viscous creep in mylonitic rocks can intrinsically produce a dynamic permeability, called creep cavitation (cf. Mancktelow et al., 1998; Herwegh and Jenni, 2001; Dimanov et al., 2007; Fusseis et al., 2009). The most well known formulation of this paradigm is the Generalised Thermodynamic model (cf. Hobbs et al., 2011) known as the dynamic granular fluid pump (Fusseis et al., 2009). While much of this paradigm remains to be tested, the notion that mylonites generate self sustaining and dynamic pathways for mass transport is radical and consequential for the interpretation of how deep shear zones behave during deformation.

The dynamic permeability is proposed to be created and sustained through the opening and closure of syn-kinematic pores, called creep cavities, by viscous grain boundary sliding during creep (e.g. Herwegh and Jenni, 2001; Dimanov et al., 2007; Fusseis et al., 2009). In recent years the paradigm has gained traction with many more contributions interpreting the presence of, or appealing to creep cavities in natural samples (e.g. Gilgannon et al., 2017; Précigout et al., 2017; Lopez-Sanchez and Llana-Fúnez, 2018; Giuntoli et al., 2020). With the most notable claims involving creep cavities being that in polymineralic viscous shear zones their formation establishes an advective mass transport pump (Fusseis et al., 2009; Menegon et al., 2015; Précigout et al., 2019), it aids melt migration (Závada et al., 2007; Spiess et al., 2012) and has even been speculated to nucleate earthquakes (Shigematsu et al., 2004; Dimanov et al., 2007; Rybacki et al., 2008; Verberne et al., 2017; Chen et al., 2020). However, much of the most convincing supporting evidence currently available is limited to deformation experiments on fabricated geo-materials and is generally restricted to grain-scale observations. Hence it has been difficult to evaluate if this phenomenon is extensive and relevant at the material scale for natural samples and, moreover, if it is applicable to natural deformations in deep shear zones.

In this contribution we provide unambiguous experimental evidence in a natural starting material that supports, and extends, the paradigm concerning the role of creep cavities in shear zones. We present quantitative results showing that creep cavities are a spatially significant feature of viscous deformation, being generated in periodic sheets throughout the samples. Our analyses are intentionally made over large areas of the experimentally deformed samples in order to contextualise and understand the role of creep cavities at a scale more comparable to those where macroscopic material descriptions are unusually made. We argue that our results warrant a reappraisal of the community's perception of how viscous deformation proceeds with time in rocks and suggest that the general model for viscous shear zones should be updated to include creep cavitation. A key consequence of this would be that the energetics of the deforming system become the keystone of our perspective rather than the mechanics.

## 2 New results from classical experiments

To make this argument, we have revisited the microstructures of a set of classical shear zone formation experiments performed on Carrara marble (Barnhoorn et al., 2004). The torsion experiments were run at a high homologous temperature ($T$ = 1000 K, $T_h$ = 0.6) with confining pressure ($p$ = 300 MPa) at constant twist rates. Samples were deformed to large shear strains and recorded the dynamic transformation of undeformed, homogeneous, coarse-grained marbles into fine-grained ultramylonites. The experiments demonstrated that microstructural change by dynamic recrystallisation was concurrent with mechanical weakening and the development of a strong crystallographic preferred orientation. More recently, it was shown that these experiments contain creep cavities and that the pores emerged with, and because of, grain-size reduction by sub-grain rotation recrystallisation (Gilgannon et al., 2020). In this contribution we expand on these observations and present new results that quantify and contextualise the development of porosity inside of an evolving viscous shear zone.

Please refer to the appendix for details of the methods used in the following results.

### 2.1 Porosity evolution with mylonitisation

At very low shear strains, and before any dynamic recrystallisation, pores decorate grain boundaries and appear as trails through large grains (d ≈ 200 $\mu$m, fig. 1a). These pores are likely fluid inclusions trapped in and around the original grains (Covey-crump, 1997) (the pore density map in fig. 1b reflects this by highlighting the outlines of the initial grain-size). In the experiment run to a shear strain of 5, which is in the midst of significant microstructural adjustment, the porosity has a clearly different character. The pores appear at the triple junctions of small recrystallised grains (d ≈ 10 $\mu$m) and in some cases are filled with new precipitates (fig. 1c). The pore density map of this experiment highlights that pores appear in clusters and that these clusters repeat across a large area with a systematic orientation (fig. 1d). Once the microstructure is fully recrystallised ($\gamma$ = 10.6), and has reached a microstructural steady state, the porosity forms elongated sheets (fig. 1e). The density map of this experiment reveals that this porosity has become more spatially extensive and also shows a systematic orientation (fig. 1f). These pore sheets contain new precipitates of mica, Mg-calcite and pyrite, implying that the sheets are permeable and act as mass transfer pathways (fig. 2). When quantified, it appears that the porosity values before and after dynamic recrystallisation are similar but as strain increases the porosity increases by an order of magnitude (table 1).

**Table 1.** Sample porosity

| Sample | $\gamma_{max}$ | Porosity (%) |
|--------|------|--------------|
| PO344 | 0.4 | 0.29 |
| PO422 | 5.0 | 0.20 |
| PO265 | 10.6 | 1.15 |

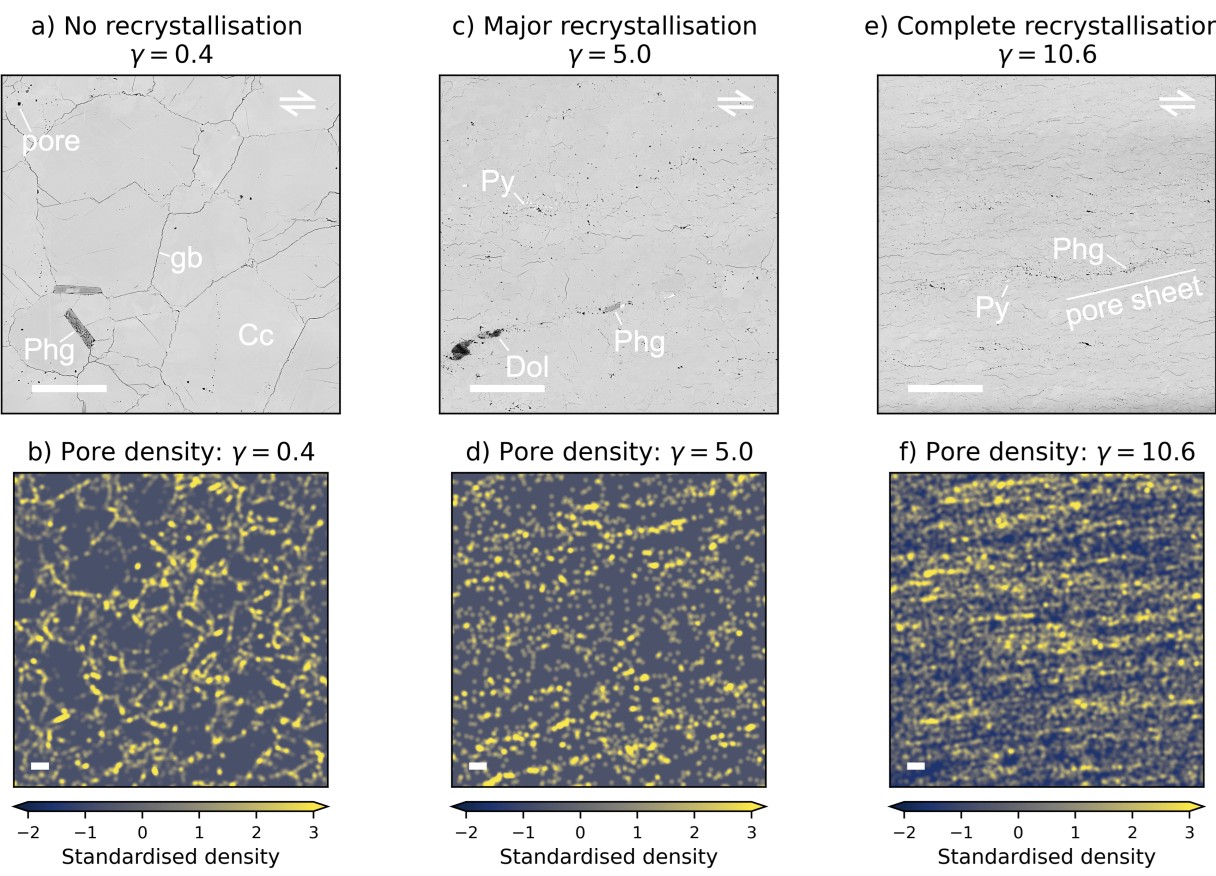

**Figure 1.** Microstructure and pore density in samples with increasing strain. The samples document the production of a mylonite through dynamic recrystallisation. Panels a,c and e are backscatter electron images while b, d and f are pore density maps. In all images the white scale bar is 100 $\mu m$. The pointed ends of the colour bars refer to the fact that some data values are larger than the max and min of the colour map. (Py = pyrite, Dol = dolomite, Phg = phengite, Cc = calcite, gb = grain boundary).

## 2.2  2D continuous wavelet analysis of pore sheets

We quantify the spatial extent and character of these permeable pore sheets with 2D continuous wavelet analysis. In particular, we use the fully-anisotropic 2D Morlet wavelet (Neupauer and Powell, 2005) to identify features in the pore density maps and expand a 1D scheme of feature significance testing used in climate sciences (Torrence and Compo, 1998) to 2D to filter for noise in the data. Furthermore, by implementing a 2D (pseudo) cone of influence we exclude boundary effects of the analysis at large wavelengths. For details of the wavelet analysis see the Methods section. Fundamentally, wavelet analysis can be thought of as a filter that highlights where the analysed data interacts with the wavelet most strongly. By varying the size and orientation ($\lambda$ and $\theta$ in fig. 3a) of the Morlet wavelet one can isolate significant features in the data and gain quantitative information about

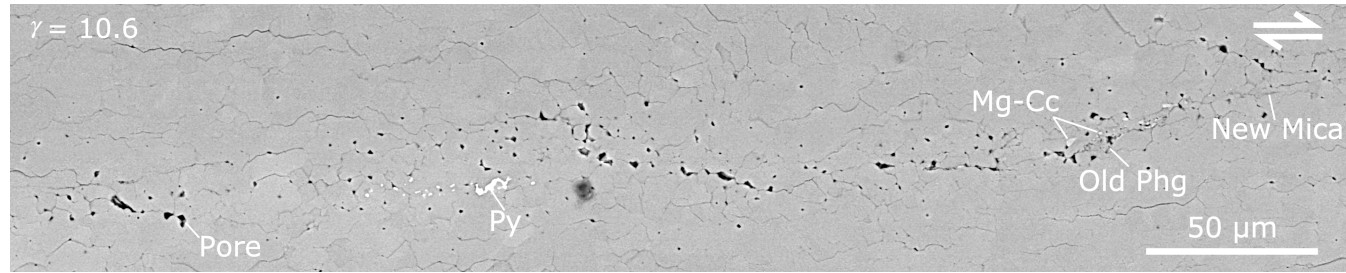

**Figure 2.** A more detailed view of the pore sheet labelled in figure 1e. Please find supporting spectra from Energy Dispersive Spectroscopy (EDS) in the appendix for the small precipitates. (Py = pyrite, Phg = phengite, Mg-Cc = magnesium calcite).

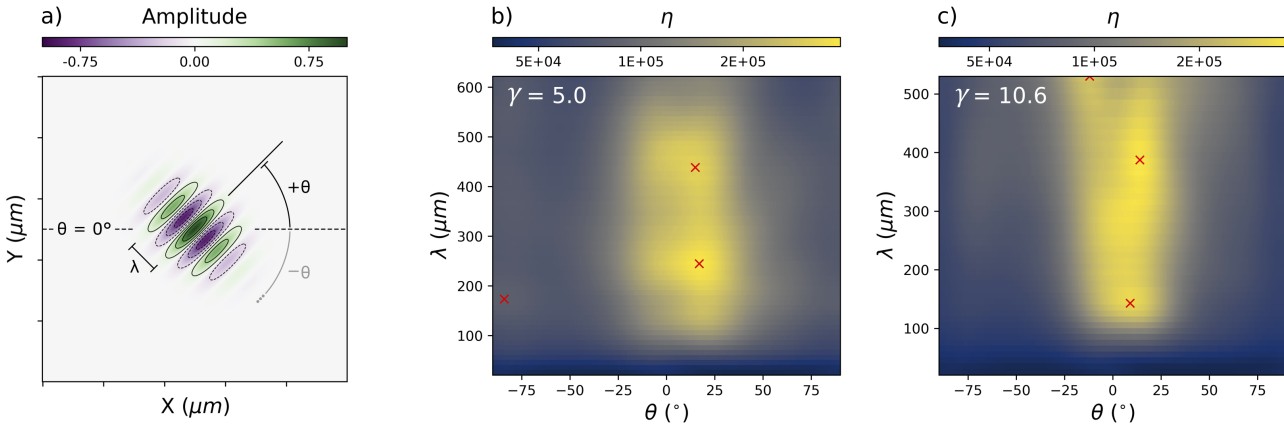

**Figure 3.** Wavelet analysis of partly ($\gamma$ = 5.0) and fully ($\gamma$ = 10.6) recrystallised samples. Fig. 3a show's a generic 2D Morlet wavelet. The wavelet analysis is conducted by considering the wavelet's interaction with the porosity density maps at each spatial position. This is repeated for different orientations ($\theta$) and wavelengths ($\lambda$). Figures 2b and c visualises the wavelet analysis results for the two samples. Peaks in $\eta$ represent the largest interaction with the wavelet (see section A7 for details). Peaks are identified by local extremes in $\eta$ and marked with red crosses. We note that two local extremes are not discussed in the text. This is because one was very close to the *sensible limit* of the analysis ($\theta$ = -12°, $\lambda$ = 530 $\mu m$) defined in appendix section A6 and the other did not correlate with any microstructural features ($\theta$ = -84°, $\lambda$ = 173 $\mu m$).

them, including orientation, dimension and any spatial frequency.

Wavelet analysis reveals that, in both the partly and fully recrystallised samples, porosity is highly ordered with a strong periodicity and anisotropy. Both samples show two dominant modes of porosity distribution (fig. 3b and c). While the sample is only partly recrystallised, porosity is preferentially oriented at 17 and 15 degrees (measured antithetically in relation to the shear plane, see fig. 3a) with wavelengths of $\sim$ 240 and $\sim$ 440 $\mu m$ respectively (fig. 3b and fig. 4a, b and c). This is both

contrasted and complemented by the modes found in the fully recrystallised experiment, where the anisotropy is oriented at 9

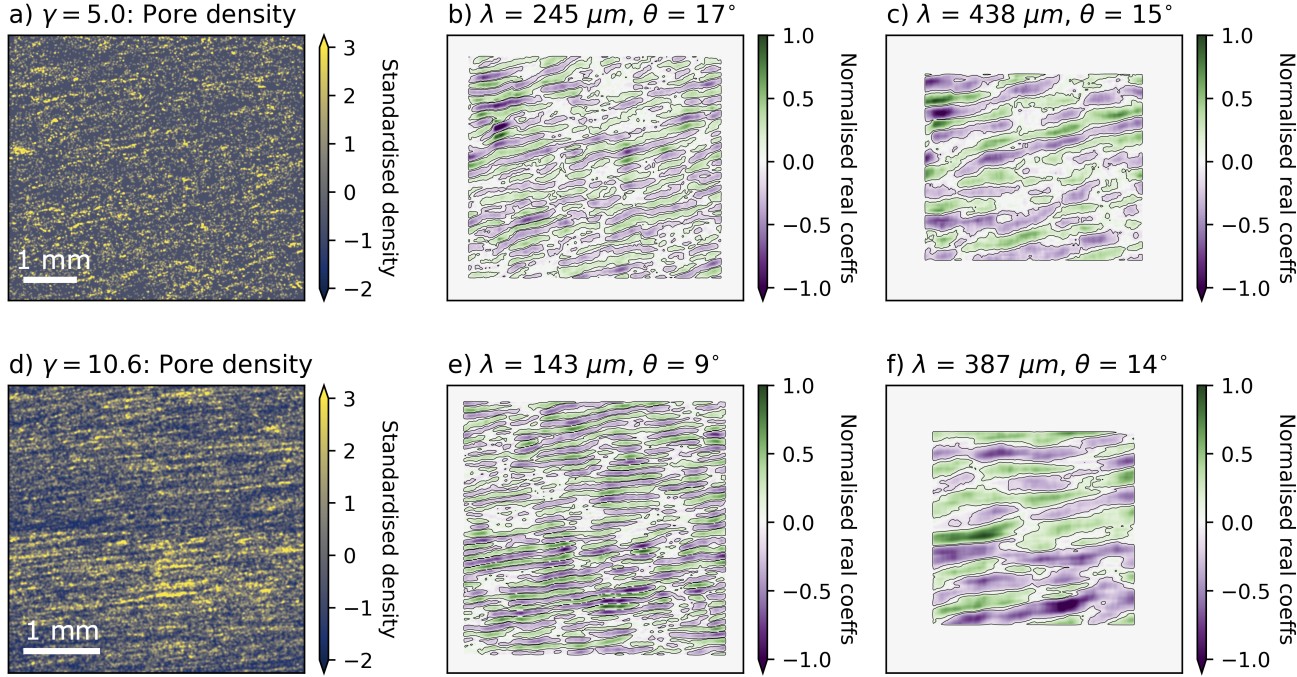

**Figure 4.** Visualisation of the results of the wavelet convolution with the pore density maps. The anisotropy identified by the dominant peaks in fig. 3 are shown for the partly (figs. 4a-c) and fully (figs. 4d-f) recrystallised samples. For each convolution at each wavelength analysed areas are defined for where edge effects may occur and data here is removed, this results in the white areas in the edge of figs. 4b-c and e-f (see section A6 for details). As before, the pointed ends of the colour bars refer to the fact that some data values are larger than the max and min of the colour map.

and 14 degrees with wavelengths of $\sim 140$ and $\sim 390\ \mu m$, respectively (fig. 3c and fig. 4d, e and f). Interestingly the longer wavelength porosity features in both samples share similar orientations and spacing ($\Delta\ 1°$, $\Delta\ 50\ \mu m$). As wavelet analysis does not require features to be periodic to be identified, the periodicity is a result and not an artefact of the analysis.

## 3 Discussion

These results provide an unambiguous foundation for discussing the community's perception of how viscous deformation proceeds with time and more generally the role of viscous deformation in the conceptual shear zone model. We claim this because our results show that a mylonitic shear zone deforming viscously can spontaneously develop highly anisotropic and periodic porous domains. This is not something that is expected within the prevailing paradigm for the deformation of rocks at high temperatures and pressures. For this reason it is important for us to reconsider the role of mylonites during geochemical,
geophysical and hydrological processes in the lithosphere.

## 3.1 How mylonites could focus mass transport

Firstly, we suggest that the presence of periodic, porous sheets in natural shear zones would act to focus fluid during active deformation. Geochemical studies have proposed that the enrichment or depletion of elements in purely viscous shear zones must reflect syn-deformational fluid migration (e.g. Carter and Dworkin, 1990; Selverstone et al., 1991). Our results provide an experimental insight into aspects of the syn-kinematic pore network that likely facilitates this fluid transport in natural mylonites. The fluid phase in our experiments is not constrained but likely some mix of $CO_2$, $H_2O$ and Ar that is both inherited from fluid inclusions of unknown compositions in the starting material, from decarbonation of the dolomite present (cf. Delle Piane et al., 2008), the breakdown of some, but not all, phengite minerals (cf. Mariani et al., 2006) and Ar that has likely diffused into the sample from the confining medium. While it is unclear what the exact composition of the fluid was, the presence of many newly precipitated minerals in pores, and across pore clusters, is evidence that mass was mobile and redistributed during the deformation. Thus if extrapolated to a natural deformation where the chemical system may be more open, mass transport through this porous network could lead to mass gain or loss in the mylonite rather than just redistribution. Additionally, our results validate the prediction of pore sheets in the dynamic granular fluid pump model (Fusseis et al., 2009) and extend it to show that pore sheets can develop spontaneously in homogenous rocks, with a periodic and oriented character. Curiously, our results also seem to suggest that porous domains develop within zones of stable orientation and, possibly, wavelength (approx. $15°$ from the shear zone boundary with a wavelength of $400 \ \mu m$). This is consequential because it implies that the emergence of porous sheets is determined by some bulk material characteristic (for example, like the elastic moduli) and not by the positions of any initial heterogeneities hosted within the starting material (akin to grain-size variations). This speculation about the role a bulk material characteristic controlling the location of pore sheet formation is possibly supported by other experimental observations that the regular spacing of creep fractures from the coalescence of pore sheets changed with temperature (see fig. 16 in Dimanov et al. (2007)). What exactly governs the appearance and location of these apparently stably oriented and spaced microstructural adjustments is a clear candidate for important future research as it hints at a challenge to the widely cited role of material inhomogeneity in determining the location of deformationally induced transformations and fluid pathways often cited in geological studies (e.g. Goncalves et al., 2016; Fossen and Cavalcante, 2017; Giuntoli et al., 2020).

## 3.2 Does a porous anisotropy affect the mechanics of a mylonite?

Two questions naturally arise from our results: (1) how could the presence of periodic porous sheets affect the mechanical behaviour of mylonites in the deep lithosphere?; and (2) why did the emergence of a spatially extensive and anisotropic porosity not affect the viscous mechanical state recorded in the original experiments of Barnhoorn et al. (2004)? Answering these questions is not trivial but there is some ground to be gained by considering the broader Generalised Thermodynamic literature behind the paradigm concerning creep cavities and contrasting our results to other experiments where creep cavities have been observed to have an mechanical impact.

### 3.2.1 What effect are creep cavities expected to have in a Generalised Thermodynamic model?

In the dynamic granular fluid pump model, creep cavities are predicted to emerge as one of several dissipative processes that act to bring the reacting and deforming rock mass into a thermodynamic stationary state (Fusseis et al., 2009). That is to say that the chief concern of the model is that of the energetics of the system with a focus on the rate of entropy production. Theoretically this means no process is a priori excluded from activating and the most efficient combination of processes that use and store energy act in congress to produce a thermodynamic stationary state (Fusseis et al., 2009; Regenauer-Lieb et al., 2009, 2015): the system is neither accelerating or decelerating in a dissipative sense but is in some kind of steady state. In this model, many factors play a role in determining whether or not creep cavities will affect the mechanical state of the deforming body.

One of these factors that may be pertinent to our experiments is the boundary conditions for deformation. It is known from various types of modelling that different boundary conditions, i.e. constant force vs constant velocity, can promote or inhibit material instability and localisation (e.g. Fressengeas and Molinari, 1987; Cherukuri and Shawki, 1995; Paterson, 2007). In the case of constant velocity boundary conditions, like those applied in our torsion experiments, localisation is not expected to occur (e.g. Fressengeas and Molinari, 1987; Paterson, 2007). Indeed, from a Generalised Thermodynamic perspective, when the boundary conditions are set to a constant thermodynamic flux, i.e. a constant velocity (like the experiments we revisit) (cf. Regenauer-Lieb et al., 2014), the dissipative conditions are fixed and the material is forced to meet them through the activation of as many dissipative micro-mechanisms, at as many positions in the rock, as necessary (cf. Veveakis and Regenauer-Lieb, 2015; Guével et al., 2019). Conducting a deformation experiment in this fashion means that the onset of a localising instability can be missed because the rock is not allowed to incrementally adjust to an incrementally applied energy input (e.g. Peters et al., 2016). This does not prohibit the possibility of local microstructural differences developing but it does mean that, at the scale of the material a distributed deformation can remain favourable despite a heterogeneous microstructure. This observation of a stable but heterogeneous microstructure suggests that the size of the heterogeneities produced are not sufficiently large to impose further localisation. In the work of Shawki (1994) it was shown that thermal perturbations below a critical wavelength would not impose localisation during constant velocity boundary conditions. If the variation in creep cavity domains reflect different amounts of work being dissipated locally, and hence heat being produced, then one can see that the shorter wavelength porous domains, which reflects the size of actual pore sheets (fig. 2), are below the size of the thermal heterogeneities that were found to impose localisation in the constant velocity models (see fig. 5 in Shawki, 1994). Thus, from a Generalised Thermodynamic perspective, the boundary conditions of our experiments may, in part, explain why we do not see an obvious mechanical effect of the porosity with ongoing straining: once the steady state microstructure is attained, the sample is in a thermodynamic stationary state for the imposed boundary velocity that favours a distributed deformation.

A constant force boundary condition (also known as a constant thermodynamic force boundary condition (cf. Regenauer-Lieb et al., 2014; Veveakis and Regenauer-Lieb, 2015)) is predicted to produce instability and localisation in rock deformation

at high homologous temperatures (e.g. Fressengeas and Molinari, 1987; Paterson, 2007). It is often assumed that plate boundaries in nature will be under such a boundary condition (cf. Alevizos et al., 2014) and in this instance the presence of anisotropic domains of porosity may have a different impact than in our experiments. For example in experiments, not dissimilar in geometry to our own, run on olivine, it was found that localisation did indeed occur at constant force conditions and not for constant velocity (Hansen et al., 2012). At a constant force this localisation was expressed both in the microstructural adjustments (with the development of an oriented foliation and domains of varying grain-size) and in the mechanical behaviour of the olivine aggregates (noted by a continual weakening of the samples beyond a shear strain of 0.5). If we for a moment speculate on how our experiments may have proceeded under constant force boundary conditions it could be that porous domains emerge with some similar modes of periodicity to those observed under a constant velocity but in this case they might provide the sites for some kind of instability. For example, in our hypothetical case, pore sheets may aid in establishing features like the high frequency foliation and lower frequency domains of grain-size variation observed by Hansen et al. (2012) (e.g. figs. 5c and d and fig. 9a in Hansen et al. (2012)). Of course our speculation is only that and this line of argument requires new experimental testing that is beyond the scope of our revisiting of the classical experiments of Barnhoorn et al. (2004). What it does highlight is that viscous deformation in mylonites requires more research to understand exactly when and where a periodic occurrence of creep cavities could have a mechanical impact.

### 3.2.2 A comparison to other experiments that developed domains of creep cavities

In this context, there are four other experimental works in which creep cavities were documented to develop that are worth comparing to our results. All of these experiments were run in torsion with constant twist rates (constant thermodynamic flux boundary conditions) at high confining pressures and homologous temperatures on synthetic dolomite (Delle Piane et al., 2008), synthetic gabbro (Dimanov et al., 2007) and synthetic anorthite aggregates (Rybacki et al., 2008, 2010). All experiments developed creep cavities in oriented and spaced domains during linear viscous flow. In the gabbroic and some of the anorthitic samples, these porous domains became sites for the generation of instabilities known as creep fractures (Dimanov et al., 2007; Rybacki et al., 2008, 2010). When one compares the four experimental sets to our samples and one another, it is clear there are many differences and similarities. Firstly, the starting materials are all compositionally different and have various initial mean grain sizes, grain shapes and grain size distributions. Secondly, all four of these experiments use fabricated samples in comparison to our natural Carrara marble samples. Thirdly, when domains of creep cavities did emerge in the four experiments, some produced bands that were broadly a mirrored orientation to our results around the shear plane (see fig. 14 and 16 in Dimanov et al. (2007); fig. 1 in Rybacki et al. (2008); fig. 6 in Rybacki et al. (2010); and fig. 2 in Spiess et al. (2012)) with others being similarly oriented to our results (see fig. 8a in Delle Piane et al. (2008) and fig. 5 in Rybacki et al. (2010)). Lastly, for the gabbroic and anorthitic experiments these oppositely oriented porous domains were reported to evolve into fractures while those domains of a similar orientation to our results did so less or not at all (Dimanov et al., 2007; Rybacki et al., 2008, 2010), and never in the dolomite experiments where linear viscous flow was maintained (Delle Piane et al., 2008). It is not clear what critical condition leads some to fracture and others to not: for example, is it the local pore density, differences in pore fluid pressure or the widths of the porous domains that controls if a fracture develops? While it is hard to draw any categorical

conclusions from the comparison of these experiments, it is noteworthy that in each experimental case the mechanical data recorded a viscous deformation, regardless of whether fracture instabilities occurred or not. This point draws attention to a conclusion already made in the seminal work of Dimanov et al. (2007), " *[c]learly, the 'microstructural state' is not obviously representative of the 'mechanical state'...* ". If this holds true for mylonites in nature then it opens an ambiguity over how the deformation of a mylonite will proceed with time: will it fracture, or will it flow?

### 3.2.3    Is a flow law enough to describe a mylonite?

Our results, and those of the four other experiments described above, suggest that mylonites of various compositions develop complicated microstructures that for an unknown set of critical conditions can facilitate a spontaneous mechanical change from flow to fracture. While there are many ways to incorporate history-dependence into flow laws that account for some microstructural change (cf. Renner and Evans, 2002; Barnhoorn et al., 2004; Evans, 2005) it does not seem that this kind of rate equation would capture the differences between ours and the four other experiments described. For example, a flow law that integrated strain (e.g. Hansen et al., 2012) would not be able to account for why some of the five experiments fractured at shear strains below  5 (e.g. Dimanov et al., 2007; Rybacki et al., 2008, 2010) with others flowing up to a shear strain of 50 with no fractures developing (Barnhoorn et al., 2004). This observation also seems to highlight a limitation to the use of empirical relationships that link strain or time to failure by creep fracture (e.g. Rybacki et al., 2008). Taken together, we suggest that this potential disconnection between the microstructure and mechanics of a mylonite places a need for the use of more complex physics to describe how a shear zone may deform with time.

This point complements the fact that the dynamic granular fluid pump model (Fusseis et al., 2009), which our work tests aspects of, requires the consideration of energy, mass and momentum balance alongside rate equations for irreversible physical deformation (like plastic or viscous flow laws) and both reversible and irreversible chemical processes (Fusseis et al., 2009; Regenauer-Lieb et al., 2009, 2015). These different dissipative processes act at different diffusive length scales and are predicted to account for the geometry and temporal variation of several phenomena that all act synchronously during a deformation (cf. Regenauer-Lieb et al., 2015). In this perspective a flow law(s) is necessary but not sufficient to fully describe a deformation, with the balancing of the energy equation being of chief importance. Said another way, many thermally activated rate equations for physical processes may compete to dissipate energy and collectively produce a bulk mechanical diffusivity (Veveakis and Regenauer-Lieb, 2015). While our results cannot speak to all of these claims, they do show that there is some validity to the predictions of this newer paradigm, namely the emergence of pore sheets. This generally adds weight to older discussions about the possible need for a more complex view of deformation in mylonites (Evans, 2005; Mancktelow, 2006; Dimanov et al., 2007) and suggests further testing is needed of the Generalised Thermodynamic ideas behind the paradigm concerning creep cavities.

### 3.3 Some consequences of incorporating creep cavities into the conceptual shear zone model

There are several enigmatic observations that are not well accounted for in the current conceptual model for lithospheric shear zones. To name a few: there is field evidence of frictional melting in the deep crust (e.g. Hobbs et al., 1986); the intrusion of dykes during upper amphibolite facies conditions (Weinberg and Regenauer-Lieb, 2010) and the fact that some geophysical data suggest that slow earthquake phenomena can occur at depths below the seismogenic zone (Wang and Tréhu, 2016). If the new evidence that we present, that mylonites can develop periodic porous domains during a viscous deformation, is incorporated into our conceptual model for lithospheric shear zones many new possible explanations emerge for otherwise hard to explain observations.

In the case of dyke intrusion at high grade conditions, the work of Weinberg and Regenauer-Lieb (2010) infact already invoked creep cavities and their coalescence into creep fractures as the responsible mechanism for allowing dyking to occur. While our results do not show the development of creep fractures, they forward the speculative argument of Weinberg and Regenauer-Lieb (2010) that creep cavities will occur during ductile shearing in rocks and could be interpreted to add weight to the discussion of Spiess et al. (2012) about the grain-scale role of creep cavities in promoting melt segregation and flow. When this is considered alongside the body other seminal experimental work on partially molten rocks (e.g. Kohlstedt and Holtzman, 2009), it becomes clear that tests need to be devised to distinguish between the different theories of melt segregation and migration (e.g. compaction length vs. sheets of creep cavities).

Additionally, recent experiments on calcite gouges made observations of creep cavities and argued that their formation allowed the gouge to transition from flow to friction (Chen et al., 2020). While these were lower temperature experiments than our own, they reinforce the notion that rocks could spontaneously transition from a viscous rheology to another mechanical state. This is relevant for observations of deep seated frictional melting where the presence of a porous anisotropy in mylonites may facilitate changes to some kind of granular or frictional mechanical mode that is otherwise unexpected. This would compliment earlier work that suggested that the mechanical behaviour of mylonites may be more pressure dependent that generally assumed (Mancktelow, 2006) and the emergence of creep cavities with high strains would facilitate this.

In the case of quartz and calcite-dominated crustal-scale shear zones there often exists a peculiar relationship between viscous strain localization in ultramylonites, fracturing and precipitation of synkinematic veins/fluid flux within these ultramylonites (e.g. Badertscher and Burkhard, 2000; Herwegh and Kunze, 2002; Herwegh et al., 2005; Haertel et al., 2013; Poulet et al., 2014; Tannock et al., 2020) . In the wake of our results, it is tempting to propose that this syn-kinematic veining may be related to an interplay of fluid transport and the instability of the anisotropic porous domains. This is especially so for cases like the deeper portions of the basal shear zones of the nappes of the Helevtic Alps, where veining is observed to broadly increase with proximity to ultramylonitic shear zones (cf. Herwegh and Kunze, 2002) that are expected to be purely viscous in

our current mechanical paradigm of the crust.

Moreover, any instability of pore sheets in natural plate boundaries may factor into explaining how both ambient and tele-seismically triggered tremors can occur at depths below the seismogenic zone (Wang and Tréhu, 2016). This could place sheets of creep cavities alongside brittle fracturing and non-isochoric chemical reactions as the potential nuclei of slow earthquake phenomena. As the emergence of creep cavities was linked to dynamic recrystallisation (Gilgannon et al., 2020), a process expected throughout the lithosphere, the porous anisotropy presented here would allow slow earthquake phenomena to occur across a range of metamorphic conditions and mineralogical compositions (Peacock, 2009).

## 4  Conclusions

In summary, the current paradigm of viscosity that is borrowed from fluids is not a completely adequate analogy for solid geomaterials. We claim this because we observe that during the formation of a mylonite, that was recorded to possess viscous mechanical properties, a heterogeneous microstructure of periodic porous domains emerged. The current paradigm of viscosity in rocks does not expect this to occur and we have discussed how this observation of pore sheet formation during viscous deformation is seen in at least four other experiments of differing compositions. The porous anisotropy we observe likely has a role to play in both the transport of mass and the mechanics of the lithosphere. On this basis, we advocate for an update to the current concept of viscosity at high temperatures and pressures in rocks to include the periodic porous anisotropy we have presented. Our discussion has explored some of the possible consequences of changing our paradigm and moving forward these speculations should be further tested. As the viscosity of solids is a cornerstone of Geoscience, our results have farther reaching implications than the conceptual shear zone model and may even be relevant for other scenarios where solid state deformation is modelled with viscous rheologies, like glacial flow (e.g. Egholm et al., 2011) and tectonics on exoplanets (e.g. Noack and Breuer, 2014).

*Code and data availability.*  Available from James Gilgannon.

## Appendix A:  Methods

The results of the main manuscript come from the investigation of 3 experimental samples (see table A1). Each sample was imaged using scanning electron microscopy, segmented and analysed. In the following the data acquisition, processing and analysis used will be outlined. For a description of the starting material or the original experimental procedure please refer to Barnhoorn et al. (2004) and Gilgannon et al. (2020).

## A1    Acquisition of large backscatter electron mosaics

Three large BSE maps were acquired on a Zeiss Evo 50 SEM with a QBSD semiconductor electron detector (acceleration voltage = 15 kV; beam current $\approx$ 500 pA). In each case the maps were stitched together by the Zeiss software Multiscan. The pixel dimensions and scales are listed in table A2.

**Table A1.** Experimental samples revisited

| Sample | $\dot{\gamma}$ | $\gamma_{max}$ | Amount of recrystallisation |
|--------|------|------|------------------|
| PO344 | 3 x 10$^{-4}$ | 0.4 | None |
| PO422 | 3 x 10$^{-4}$ | 5.0 | Major[a] |
| PO265 | 2 x 10$^{-3}$ | 10.6 | Complete[b] |

[a] 65-90 % as classified by Barnhoorn et al. (2004)

[b] 90-100 % as classified by Barnhoorn et al. (2004)

**Table A2.** Dimensions and resolutions of mosaics

| Sample | $\gamma_{max}$ | Pixel dimensions | Scale (px : $\mu m$) |
|--------|------|------------------|----------------|
| PO344 | 0.4 | 15000 x 13944 | 1 : 0.36 |
| PO422 | 5.0 | 8745 x 7392 | 1 : 0.55 |
| PO265 | 10.6 | 21127 x 20494 | 1 : 0.16 |

## A2    Segmentation for porosity

We used the segmentation, labelling and filtering work flow described in Gilgannon et al. (2020) for open pore space. In this work flow grain boundaries must be filtered for by using each labelled feature's aspect ratio. Specifically, the data is filtered to remove features with aspect ratios greater than 4. Figure A1 shows all features initially labelled as porosity by the segmentation process in each sample. These are plotted for their area and perimeter, while colour coded for aspect ratio. In each data set there are two trends:

1. Features with aspect ratios $> 4$ that show area-perimeter relationships for lines of widths between 0.25-1.25 $\mu m$

2. Features with aspect ratios $< 4$ that do not show area-perimeter relations of a line

Based on this criteria, only features with aspect ratios of $< 4$ are considered as pores. The centres of mass of features meeting this criteria were then extracted and used in the kernel point density analysis.

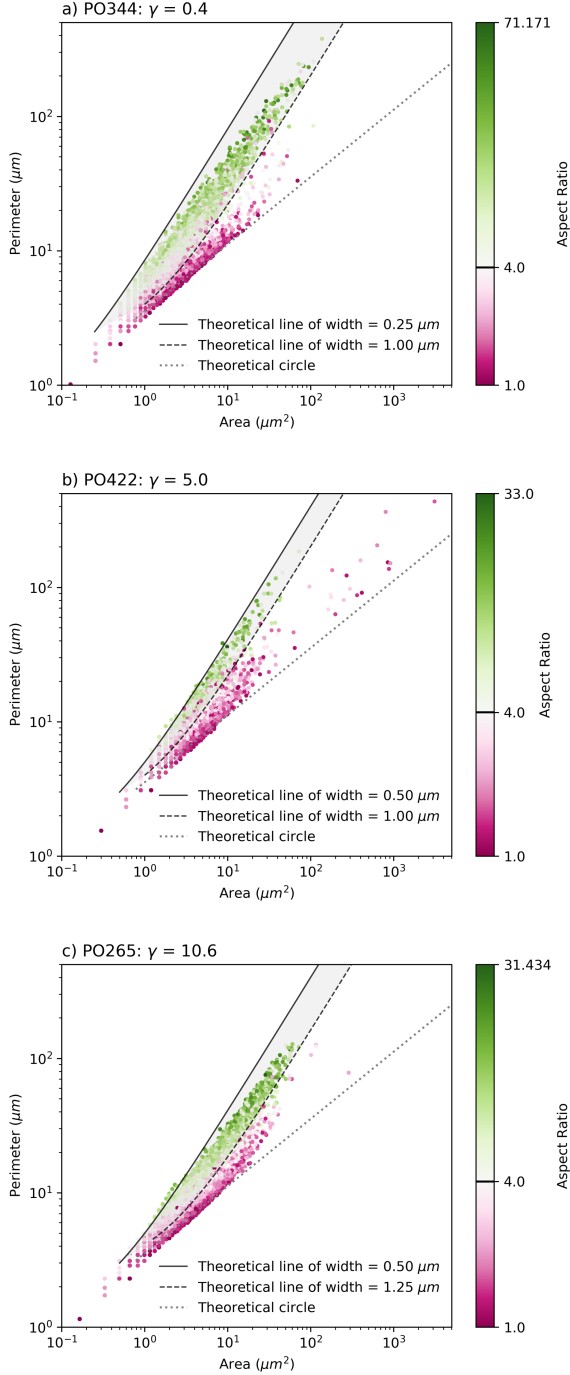

**Figure A1.** Filtering criteria for grain boundaries and pores.

**A3   Kernel density estimator maps**

At the first instance, one of the major difficulties in understanding the relationship of micron scale features across millimetres is simply visualising the problem. We utilised the kernel density (KDE) for point features function in ESRI's ArcGIS v10.1 software to overcome this issue. This has the effect of converting point data, that only tell us something about individual pores, to a map that considers the distribution of pores and, in part, their relation in space.

We manually set the output cell size and search radius to 1 $\mu m$ and 20 $\mu m$, respectively. The kernel smoothing factor was automatically calculated with reference to the population size and the extent of analysis and contoured based on a 1/4 $\sigma$ kernel. We specifically did not use the default search radius (calculated with Silverman's Rule of Thumb). Our intention here was to retain as much data as possible in the visualisation. In this way we visualised local neighbourhoods and produced an image for further analysis that had not been overly smoothed. It was these density maps that we then quantitatively analysed with 2D continuous wavelet analysis.

**A4   2D Continuous wavelet analysis**

Wavelets are highly localised waveforms that can be used to analyse signals with rising and falling intensity. Our images are such signals. Simply put, wavelets can be used to reveal the location (in space or time) and the frequency at which the most significant parts of a signal can be found. Continuous wavelet analysis is the particular wavelet-based method that we employ in this contribution.

To identify features at different frequencies the wavelet is stretched over what are known as different scales ($a$). The scales relate to the central frequency of the wavelet, which in turn can be related to the wavelength ($\lambda$):

$$\lambda = \frac{4\pi a}{k_0 + \sqrt{k_0^2 + 4}} \tag{A1}$$

where $k_0$ is the wavenumber.

In the broadest sense, a wavelet can be seen as a filter that finds peaks in an image. To do this it is shifted around the spatial domain ($\mathbf{x}$) of an image, by way of the shift parameter ($\mathbf{b}$), and this is repeated at different scales to find peaks. In this way features of different sizes can be located in space and in scale: short wavelengths highlight small features and long wavelengths larger ones. In this contribution, we utilise the fully anisotropic Morlet wavelet (Neupauer and Powell, 2005) because it also allows features of varying orientation to be identified. The wavelet is considered to be fully anisotropic because it produces

in-phase elongation along the wave vector, such that the wavelet can be rotated and maintain its anisotropy. The wavelet takes the form:

$$\Psi(\mathbf{x}, \theta, L) = e^{i\mathbf{k}_0 \cdot \mathbf{Cx}} e^{-1/2(\mathbf{Cx} \cdot \mathbf{A}^T \mathbf{ACx})} \tag{A2}$$

Where $\theta$, $L$, $\mathbf{k}_0$, $\mathbf{C}$ and $\mathbf{A}$ are the angle for the rotation matrix, ratio of anisotropy, wave vector, rotation matrix and anisotropy matrix, respectively. The non-scalar terms are given by:

$$\mathbf{k}_0 = (0, k_0), \; k_0 > 5.5 \tag{A3}$$

In this study we use $k_0 = 6.0$.

$$\mathbf{C} = \begin{bmatrix} cos\,\theta & -sin\,\theta \\ sin\,\theta & cos\,\theta \end{bmatrix} \tag{A4}$$

This rotation matrix rotates the entire wavelet by $\theta$, which is defined as positive in a counter-clockwise direction with respect to the positive x axis (see fig. 3 of the main manuscript).

$$\mathbf{A} = \begin{bmatrix} L & 0 \\ 0 & 1 \end{bmatrix} \tag{A5}$$

Where the ratio of anisotropy ($L$) is defined as the ratio of the length of the wavelet perpendicular to $\theta$ over the length parallel
to $\theta$. In this way, values of $L < 1$ represent extreme anisotropy parallel to the angle of the wavelet.

We chose to use an anisotropy ratio of $L = 1.5$ (see fig. A2). This was done because the input images are kernel density maps and the estimator used is circular. We wanted our wavelet to utilise its inherent anisotropy and angular selectivity to identify extended concentrations of the estimator that would appear as elliptical clusters of circles. We did not use an $L < 1$ as these
wavelets anisotropies are too far from the shapes expected from the features we investigated (Torrence and Compo, 1998). If, for example, we had been investigating linear features like fractures we would have used a more anisotropic wavelet shape (for example, $L = 0.5$).

To ensure that the total energy of the analysing wavelet is independent of the scale of analysis the relationship between the
wavelet (eq.A2) and the mother wavelet is:

$$\Psi_{a,\mathbf{b}}(\mathbf{x}, \theta, L) = \frac{\sqrt{L}}{a} \Psi\left(\frac{\mathbf{x} - \mathbf{b}}{a}, \theta, L\right) \tag{A6}$$

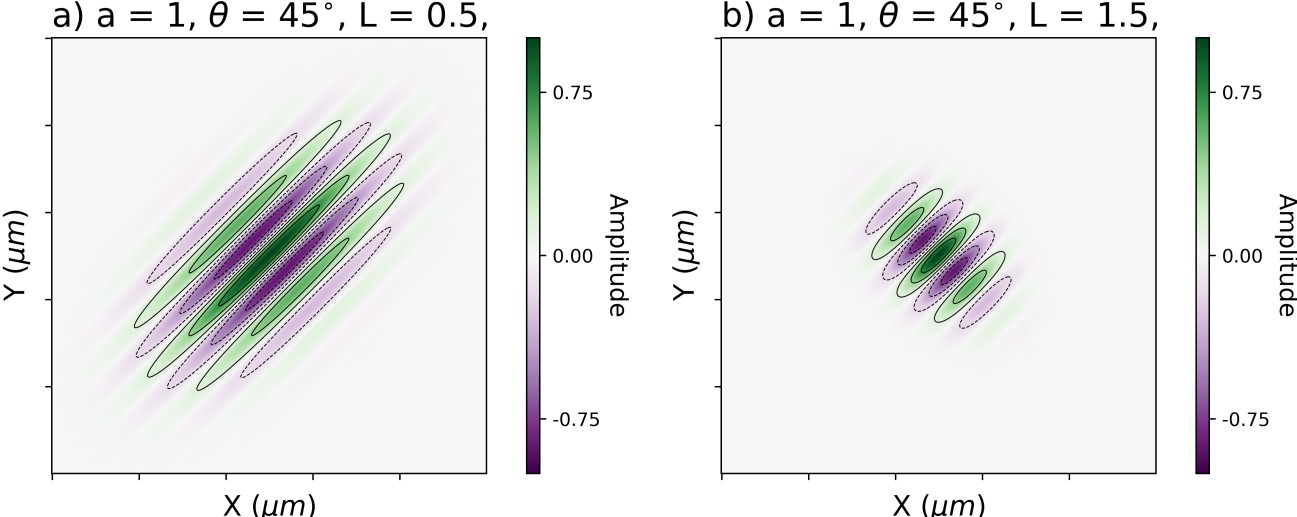

**Figure A2.** Examples of how a wavelet changes with the anisotropy ratio (L).

To be clear, we refer, above, to energy in the generalized sense of signal processing.

Our input images for wavelet analysis are 32-bit kernel density maps for the centres of mass of open pores. Each density map can be considered as an intensity function ($\mathbf{I}(\mathbf{x})$) who's magnitude limits are that of the KDE density calculation. We standardise each input image such that,

$$\mathbf{I}_{std}(\mathbf{x}) = \frac{\mathbf{I} - \mu}{\sigma} \tag{A7}$$

Where $\mu$ and $\sigma$ are the image's mean and standard deviation. This is because the best results of wavelet analysis are achieved on a zero-mean random field (Neupauer and Powell, 2005).

It is on this new standardised image that we preform the wavelet transformation. The wavelet transformation of $\mathbf{I}_{std}(\mathbf{x})$ is a convolution with the analysing wavelet:

$$\mathbf{W}_{\boldsymbol{\Psi}} f(\mathbf{b}, a, \theta, L) = \frac{\sqrt{L}}{a} \int_{-\infty}^{\infty} f(\mathbf{x}) \bar{\boldsymbol{\Psi}} \left( \frac{\mathbf{x} - \mathbf{b}}{a}, \theta, L \right) d\mathbf{x} = \frac{\sqrt{L}}{a} f(\mathbf{b}) * \bar{\boldsymbol{\Psi}} \left( -\frac{\mathbf{b}}{a}, \theta, L \right) \tag{A8}$$

Here, $*$ is the convolution and the overbar denotes the complex conjugate. The convolution is evaluated by taking the inverse fast Fourier transform of the products of the Fourier transforms of $f(\mathbf{b})$ and $\bar{\boldsymbol{\Psi}}(-\mathbf{b}/a, \theta, L)$. This wavelet and the convolution follow those outlined in Neupauer and Powell (2005).

## A5 Defining significance

As outlined above, wavelet analysis will highlight regions of an image where the wavelet and the image interact strongly. This interaction alone is not enough to say that what the wavelet highlighted is relevant when compared to any expected noise in the image. Therefore, it is important to know if the areas highlighted by the wavelet are significant. To define what is significant in the analysis we adopt the method outlined in Torrence and Compo (1998).

The general assumption of the null hypothesis is that the image analysed has some mean power spectrum ($P_k$, see equation 16 in Torrence and Compo (1998)), related to a background geophysical process(es). If the wavelet power spectra is found to be significantly above this background spectrum then the feature is *a real anomaly* and not a result of the assumed background process(es).

To test the null hypotheses, the local wavelet power spectrum at each scale (following equation 18 in Torrence and Compo (1998)) must be considered:

$$\frac{|W_\mathbf{b}(a)^2|}{\sigma^2} \implies \frac{1}{2} P_k \chi_2^2 \tag{A9}$$

Where $|W_\mathbf{b}(a)^2|$ is the local power, $\sigma^2$ is the variance, $\implies$ indicates 'is distributed as' and $\chi_2^2$ represents a chi-square distribution with two degrees of freedom. Using the relation in eq. A9 one can find how significantly the local wavelet power deviates from the background spectrum. To do this, the mean background spectrum, $P_k$ (where $k$ is the Fourier frequency), is multiplied by the 95th percentile value of $\chi_2^2$ to give a 95% confidence level. As the local wavelet power is distributed equivalently, this confidence level can be used to contour the global wavelet power ($|\mathbf{W_\Psi}^2|$). The result allows the identification of data that has a 95% chance of not being a random peak from the background spectrum (see fig. A3).

In this contribution we adopted a white noise model as our background spectra. White noise is a random signal that assumes a uniform power across frequencies. We chose this because we wish to identify when porosity density is non-random and, based on our knowledge of the active processes, we consider that any noise will be uniform across the scales of analysis. We make this assumption about the background spectra for the following reasons.

The experiments we revisit are non-localising at the sample scale and are considered as the exemplar of a sub-solidus, homogenous, viscous deformation. The prevailing assumption for such a sample being deformed is that the microstructural change will first occur where locally favourable conditions allow. For example, some poorly oriented grains may develop more

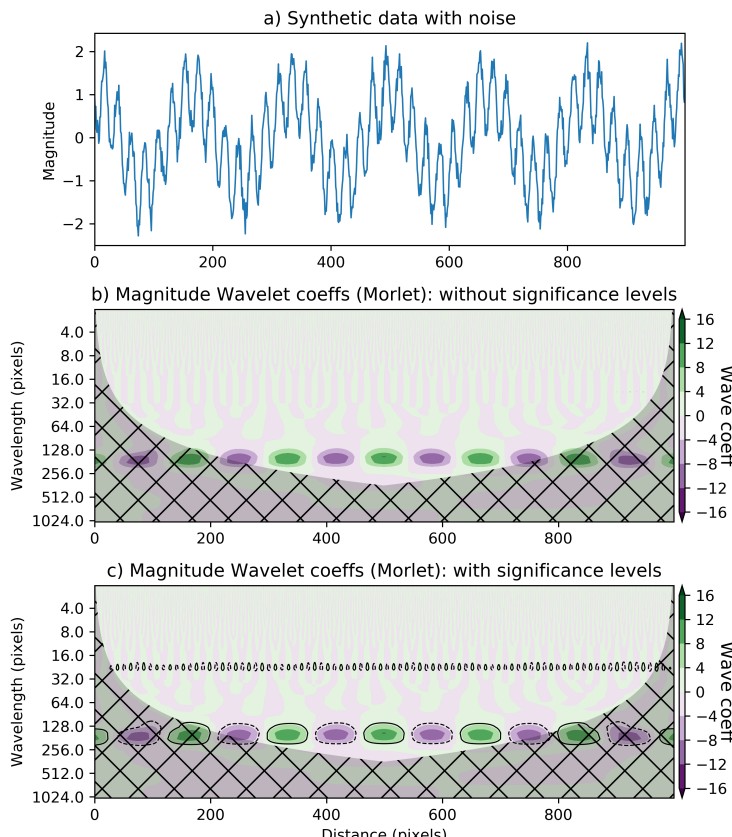

**Figure A3.** Example of why significance test is needed. a) is a 1D synthetic, noisy, signal made of 2 waves, of different wavelengths. b) and c) present the 1D wavelet transformation of (a) with a 1D Morlet wavelet: in (b) the result is only visualised, while in (c) the significance test is also visualised. In both, the hatched area contains edge effects and is delimited by a 1D cone of influence. The contouring in (c) shows domains that are within the 95% confidence level of not being white noise. The contours highlight values that are both positively and negatively larger than the white noise model. The takeaway message is that the significance test is needed to accurately identify regions of 'real anomaly'.

deformation induced defects and be prone to recrystallise earlier than other grains. The general distribution of grain orientations is determined by the starting material's texture, which in the case of Carrara marble is random (Pieri et al., 2001). Therefore, as there is not any initial anisotropy in grain orientations, it is expected that porosity will form randomly in space at favourable sites in the microstructre. Any deviation from this expectation is of interest to us. For these reasons, we use a white noise model as our background spectra and it forms the reference for testing where the porosity density is non-random.

As stated in the main text, it was shown for our experiments that creep cavities emerged with, and because of, grain size reduction by sub-grain rotation recrystallisation (Gilgannon et al., 2020). The white noise null hypothesis used supposes that

this grain size change and porosity development occurred with no preference in space or frequency. By using this as our null model we can show when the wavelet analysis produces interactions that are very unlikely to have occurred randomly, and highlights heterogeneity and anisotropy in the porosity density maps.

## A6   Defining limits of the analysis

As images have finite length and width the analysing wavelet will misinterpret these edges and produce erroneously positive results. To avoid this one may use padding but ultimately the problem will remain (Torrence and Compo, 1998). Instead, we have chosen to implement a 2D cone of influence (COI). Starting from the edge of the image, the COI defines a zone in which data will likely suffer from edge effects (see area outside of red contour in fig. A4b). The zone increases proportionally with the wavelet scales. We consider our 2D COI as a pseudo-COI because we simply project two 1D COIs across the 2D surface. We use the e-folding time defined by Torrence and Compo (1998) for their 1D Morlet wavelet, which is $\sqrt{2a}$. For both the y and x axis of the input image we can calculate the appropriate length 1D COI. Each of these 1D COIs is calculated for the set of discrete scales (a) defined for the wavelet transformation. At each scale, the y and x axis COIs are projected to produce a contour that defines the 2D COI at each scale (see fig. A4b and c).

In this way, our COI does not account for the wavelet's shape and anisotropy. While this means our COI is not the correct mathematical solution for defining where edge effects end for this 2D wavelet, it is a best first attempt at defining a limit to the analysis. We then delete data that lies within the COI. Furthermore, we define a *sensible* limit to the largest relevant scale of analysis by only considering scales which have *edge-effect-free* windows that are greater than 30% of the original image size.

## A7   Visualising wavelet results

Figure 3 of the main manuscript uses the global measure $\eta$ (Neupauer et al., 2006) to investigate peaks in the data. Here we define $\eta$ as:

$$\eta(a, \theta, L) = \int |\mathbf{W_\Psi}^2| \, d\mathbf{b} \tag{A10}$$

As we use only one value of anisotropy, $\eta$ can be visualised to reveal information about peaks in orientation and scale. To quantitatively identify peaks in $\eta$ we use the *h_maxima* function (where h = 0.03) of the Scikit-image python library (van der Walt et al., 2014).

## A8   Energy Dispersive Spectroscopy (EDS) spectra of small precipitates in pore sheet

Point analysis was used to collect spectra with Energy Dispersive Spectroscopy (EDS) of the small precipitates in the pore sheet presented in figure 2 of the main manuscript. This data is shown alongside a higher resolution BSE image taken under high vacuum on the same Zeiss Evo 50 SEM described above.

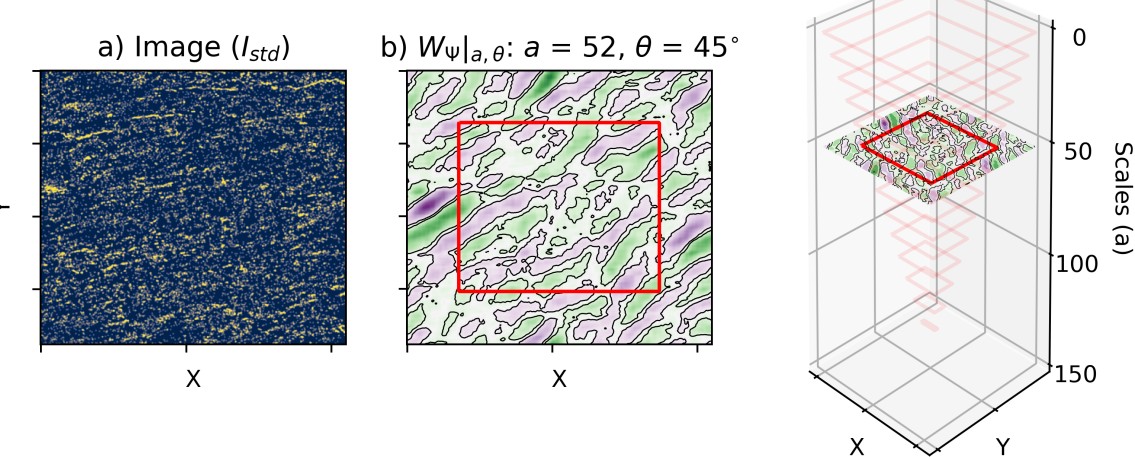

**Figure A4.** Defining the limits of the analysis across scales. For some image (a), an arbitrary wavelet transformation is visualised at an arbitrary scale and angle (b). Here the black contours enclose data that is within the 95% confidence level. Overlaying this is a red contour which delimits the zone of possible edge effects of the analysis. At this scale this is a slice of the cone of influence (COI). c) visualises the COI across scales. As the scale, and therefore the wavelength of the analysing wavelet, increase, the zone without edge effects decreases. For the purposes of demonstration, data within the COI has not been removed in this figure.

*Author contributions.* J. Gilgannon, T. Poulet, A. Berger and M. Herwegh designed the study. J. Gilgannon and M. Waldvogel implemented the wavelet method. A. Barnhoorn ran the original experiments. All authors were involved in the interpretation of the results and the writing of the final manuscript.

*Competing interests.* There are no competing interests.

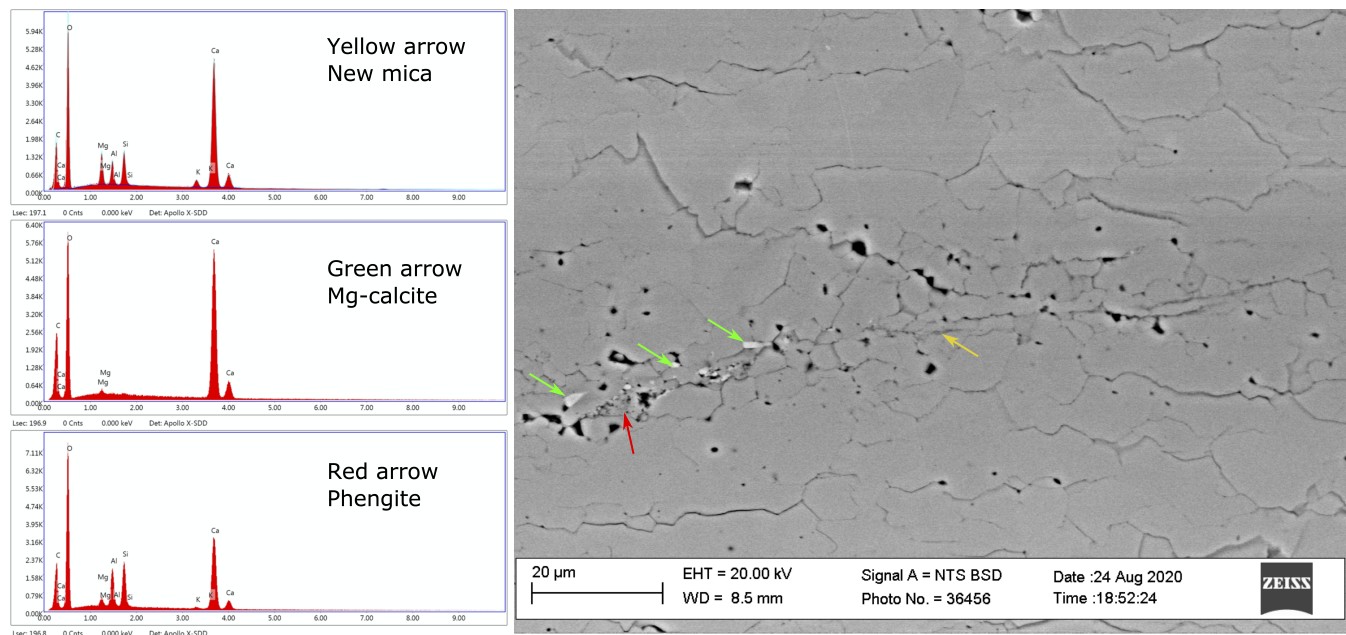

**Figure A5.** Energy Dispersive Spectroscopy (EDS) spectra of small precipitates in pore sheet shown in figure 2 of the main manuscript.

*Acknowledgements.* This work was financially supported by the Swiss National Science Foundation (SNSF; grant number 162340). We would also like to thank Klaus Regenauer-Lieb for several stimulating discussions about the role of creep cavities and rock rheology in general.

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
