# Peer review of "Experimental evidence that viscous shear zones generate periodic pore sheets"

_Solid Earth, 2020_

## Referee Comment (RC1) · Alberto Ceccato (Referee) · 19 Sep 2020

General comments: In this Short Communication, Gilgannon et al. present a series of microstructural data from experimental deformation of natural rock samples which demonstrate the systematic and spontaneous development of microstructurally–organized transient porosity, i.e. dilatant microstructural sites, during viscous deformation. This dataset and the following discussions allow the Authors to tackle the validity of the common assumption that dilatant processes and porosity development are suppressed during viscous deformation of rocks. From the title, the main discussion points to address the effects of this transient porosity on the mass transfer capability

of viscous shear zones. Then, in the main text, the effects of this porosity development are unfolded and extended also to other topics, such as deep seismicity, slow earthquakes and tremors. In a quantitative way, the Authors demonstrate that the development of transient porosity is a systematic characteristics of viscous shear zones developed in an homogeneous starting material and they demonstrate that the extent of such process is relevant to the behaviour of the whole shear zone and not only to the grain–scale interactions.

Overall, this preprint represents a substantial contribution to the scientific progress within the topics of the deformation of the solid earth, tackling quantitatively a topic which was previously only postulated or treated in a qualitative way.

The manuscript is concise, well written and straightforward. The introduction drives the reader clearly and straightforwardly to the main scientific question the Authors want to tackle in the main text. Then, data and methods are presented in a concise way and focussed on selected points necessary to the reader to follow and understand the subsequent discussions. The microstructural dataset and analysis described here are novel, scientifically sound and statistically robust, providing a sound base on which the Authors can build their discussions. The Method section is extensively unfolded and integrated in the Appendix.

However, the Discussion section suffers from just a couple of generic and vague sentences (as outlined below in the Specific Comments section), which the Authors need to recast a little bit in order to support their inferences and conclusions. I understand that the Authors are clearly speculating on some of the discussed topics, but some sentences present a very large logical jump between the presented data and the conclusions they want to support. Most of my observations and comments focusses on the speculation about pore–fracture evolution, not the mass–transfer capability, which seems to be the main topic of the communication.

I want to highlight the novelty and the importance of the quantitative approach the

Authors have adopted to tackle the topic, being a fundamental point of the present paper that makes the difference with previous publications. Moreover, I have found this Short Communication really stimulating, triggering in me a lot of new questions and interesting points of discussions. I think the present preprint need some moderate–minor revisions in order to make it suitable for publication in the Solid Earth journal. I think the paper will be of high interest for large part of the structural–experimental–geophysical community of the readers of SE.

In the followings, I provide some comments and observations, which I hope will help the Authors to improve the present preprint. Well done, congratulations!

Specific comments:

Lines 13–14: please add some references supporting this statement.

Lines 32–34: I partially disagree with these two sentences. It is true that much of the past advances on the creep cavitation subject results from experimental works and micro–scale analysis (as the cited references report); but it is also true that many other authors have evaluated the extension and occurrence of creep cavitation and related phenomena in natural polymineralic shear zones deformed at geological conditions and have tried to qualitatively extrapolate their results from the thin section–scale to crustal–scale shear zones (e.g. Giuntoli et al., 2020 SciRep https://doi.org/10.1038/s41598-020-66640-3; Preciguot et al., 2017 NatComm https://doi.org/10.1038/ncomms15736). I understand that your samples are "natural" geo–materials, but you are presenting the results of experimental deformation at lab conditions, not geological conditions. I suggest the Authors to recast these two sentences taking into consideration this observation. I would focus on two main points that, in my opinion, are the strong points and the novelty of the paper: 1) the quantitative approach to evaluate the extension of creep cavitation processes; 2) the occurrence of creep cavitation in monomineralic shear zone.

Line 40: please, specify what you have revisited of the set of classical experiments (the

microstructures of the deformed samples).

Line 55: please, specify if you have considered these grain–filled pores as "porosity" during the following density quantification or not. I guess you have considered it as "porosity", but this is not clear neither from the main text, nor from the Method section (or perhaps I missed it).

Line 89: I would replace "microstructure of . . ." with "the spatial arrangement of syn–kinematic pores". In my opinion, the definition of the effective microstructure would imply the analysis and characterization of pore typology, connection, morphology, etc. . .

Line 94: Even though I agree with the Authors about the importance of the spontaneous development of a systematic pore pattern, this statement is rather speculative and not fully supported by the data presented in the manuscript. "Bulk material properties" is rather generic and vague. What are these bulk material properties and initial heterogeneities? Grain size? Mono–polymineralic composition of the sample? (What about the experimental boundary conditions? Shear strain rate? confining pressure? Sample/Room humidity and fluid availability?) I understand that this could lead to further discussion that goes beyond the scope of this short communication, but I would like the Authors to be a little bit more specific on the term "bulk material properties" if they want to keep the sentence. On the other hand, if the Author want to work around this comment: highlight the lack of any initial heterogeneity and state that, at present, the genetic causes of this microstructure need further investigations. . .

Lines 104–106: These two sentences are rather speculative and vague. Unfortunately, it seems to me that a logical connection between the occurrence of strain invariant mechanical parameters and the possible evolution of pore sheets into creep fractures is missing. In my opinion, this paragraph suffers from two main problems: (1) there are no direct logical links between the observation of strain invariant mechanical parameters and the evolution of pore sheets into creep fractures, i.e. the maintenance of strain invariant mechanical parameters is not a sufficient conditions to prove (or just speculate) that also your pore sheets will evolve into creep fractures. (2) Even though the experiments of Dimanov et al. and Rybacki et al. have been performed under similar conditions to those reported here, there are some fundamental differences that might undermine your inference: a) different lithologies; b) occurrence of creep fractures at relatively low strains if compared to shear strains obtained in Barnhoorn et al.; c) lack of any evidence of creep fractures in the set of experiments of Barnhoorn et al. (2004). Rather than a speculation on the possible evolution of these pore sheets under "variable" natural conditions, I would suggest the Authors to discuss and compare in a more detailed manner their experimental results and inferences with those presented by Dimanov et al (2007) and Rybacki et al. (2008). Then, a speculative extrapolation to the natural conditions might be attempted. Indeed, there might be some microstructural similarities between the cited references and the data reported in the manuscript that might better support your speculation about the "possible evolution of pore sheets into creep fractures". In addition, "natural conditions are more varied" is a rather vague and generic statement. What are these variable natural conditions? Transient shear strain rates? Wet vs. dry conditions? Please specify if you want to keep the sentence.

Technical Corrections:

Figure 1: please, enhance the brightness/contrast of the BSE images OR enlarge the images (this might help the reader to instantaneously capture the porosity distribution even if the image is very small).

Figures in the supplement: There is a discrepancy between the figure call–out in the text (Figure S*) and the Figure captions (Figure A*). Please change one or the other to be consistent throughout the text.

Line 245: There's probably a reference typo [(18) ???].

---

## Referee Comment (RC2) · Lars Hansen (Referee) · 20 Sep 2020

I read with interest the paper titled "Experimental evidence that viscous shear zones generate periodic pore sheets that focus mass transport" by Gilgannon and coauthors. The paper presents microstructural analyses of marble samples deformed as part of a previously described set of laboratory experiments. Most of the authors were part of a recent paper (Gilgannon et al., GRL 2020), which also conducted a microstructural analysis of a sample from the same sample suite. In their previous paper, the authors suggest that cavities are nucleated as part of the recrystallization process. In the present paper, the authors examine the spatial distribution of those cavities and

demonstrate that the distribution becomes increasingly anisotropic with progressive deformation. The authors then conclude that these observations demonstrate that rocks cannot be described as viscous in the same manner as fluids and that constitutive models for rock viscosity are therefore inherently flawed. This content is appropriate for publication in Solid Earth.

The central analysis in the paper involves the use of wavelet convolution to characterize the spatial distribution of pores. I'm not aware of comparable implementations of wavelet analysis in microstructural studies, and I think this is a very valuable contribution to the community. The anisotropic distribution of pores is clearly demonstrated, and the caveats of the method are usefully discussed.

I do have three main comments about the interpretations of the results and the statement about constitutive models of rock viscosity. First, I'm curious about the nature of the cavities, what the pore fluid is, and how many pores are inherent in the starting material. Second, I have some concerns about the inference that porosity increases during deformation, which stems from the details of the analysis. Third, the discussion of treating rocks as viscous fluids seems like an overstatement considering the nature of evidence presented in the manuscript. I discuss these three points below, followed by minor comments indicated by line number.

–The nature of the cavities:

My question in this context is whether or not there is a pore fluid, and if so, what is its composition? One item of concern is whether the pores are filled with $CO_2$. That is, could there be some decarbonation of the calcite during initial pressurization and heating that generates porosity? Caristan, Harpin, and Evans (1981) presented evidence of calcite decarbonization during hot-pressing of calcite aggregates in a gas-medium apparatus, and it seems worth discussing whether or not something similar has occurred here. This question could be answered in some part by comparing the starting material (not shown) to the low-strain material (gamma = 0.4).

[Figure]

–The evolution of porosity and its measurement:

Considering that some porosity may be present at the onset of deformation, the fundamental question is whether or not the porosity increases with deformation. A primary conclusion of the previous work (Gilgannon et al., GRL, 2020), which analyzes one of the same samples as in this paper (PO422, gamma = 5), is that porosity increases due to pore nucleation associated with recrystallization. Indeed, the previous work presents excellent observations of pores associated with new grain boundaries, and a total porosity of ∼1%, but as far as I can tell, it is never demonstrated that deformation increases the total porosity.

The present work presents an excellent opportunity to make that demonstration since samples deformed to a variety of strains are analyzed. However, it is still not clear to me that there is a change in the total porosity. The wavelet analysis definitely demonstrates a change in the spatial distribution of pores, but as far as I understand, it does not present an increase in the porosity.

My conclusion here stems from the details of the measurement of porosity density, which is the fundamental measurement presented in the paper. As I understand the measurement procedure described in the appendix, only the centroids of identified pores are used in the construction of the porosity density maps. Thus, "porosity density" is apparently the number of pores per unit area, and this quantity may be better referred to as the "pore density". I admit I had some confusion about how "standardized density" is defined, why it is unitless, and why it can have negative values. Some clarification here would be valuable in a revised version, but for now it seems to me to represent a normalized version of the "pore density" and not the porosity (pore volume per total volume).

Unfortunately, the "pore density" does not seem as valuable as porosity in regards to interpreting the hydromechanical effects of the observed microstructural evolution. A key inference in the paper is that permeability is enhanced (and becomes anisotropic)

during deformation. The relationship between permeability and porosity is not trivial, but if the porosity were known, a back of the envelope calculation could be made to assess the potential changes in permeability expected. However, I don't think changes in permeability can be assessed qualitatively or quantitatively when using only the number of pores. Furthermore, from the data presented here, it seems entirely possible that the porosity could be constant with increasing deformation, even though the number and distribution of discrete pores evolves.

As a final note on this topic, there is a clear link to previous work on the segregation of melt during deformation of partially molten rocks (for a review, see Kohlstedt and Holtzman, AREPS, 2009). I'm surprised to not see any of that work referenced in the present manuscript. Those studies demonstrate that the (melt-filled) porosity can dynamically arrange into planar features not dissimilar to the "periodic pore sheets" described here. However, I'll emphasize that the average porosity is static in those experiments. Locally it may increase or decrease, some new pores are created, some old ones are destroyed, but the total porosity does not change. It remains to be demonstrated that a similar phenomenon is not occurring in the calcite samples analyzed here and that the overall porosity is actually increasing.

–Challenging the concept of rocks as viscous fluid:

The authors take the discussion beyond the development of a periodic array of pores to comment on whether or not rocks can be treated as viscous fluids. The paper is framed around the common treatment of rocks as viscous fluids at long timescales and suggests that that framework is flawed and a new paradigm involving creep cavitation is necessary. Emphasis is given to the transition from viscous to brittle behavior.

I suggest that these statements are overstating the case and detract from what is otherwise a useful paper about applying wavelet analysis to periodic pore distributions. My primary concern here is that the authors do not provide any evidence that the mechanical properties of the rocks investigated are modified by the formation of an anisotropic

pore distribution. The mechanical data have already been published in previous work, so if these pore sheets are significant to the rheological behavior, then the authors can demonstrate that with the mechanical data. Furthermore, if these rocks can't be described as viscous fluids because of the porosity evolution, then the authors could again demonstrate that with the mechanical data.

In addition, I'll note that viscosity is a phenomenological description, and when discussed in the context of crystalline materials, is generally only taken to apply at macroscopic scales. For example, we often discuss rocks as fluids with non-Newtonian viscosities for which the viscosity is controlled by the dynamics of dislocations at the lattice scale. There has been much work in the materials sciences over the past several decades demonstrating that, at scales well below the grain size, dislocation propagation occurs in discrete bursts (key terms are "dislocation avalanches" and "jerky flow"). This behavior is clearly not viscous, but when averaged over many crystals of many orientations, the mechanical behavior can still be described as viscous.

Key to the authors' argument is that creep cavitation can lead to brittle behavior, which "can never be predicted by flow laws commonly used to model viscous deformation." An analogous case can be seen in the fatigue of metals. Constitutive equations have existed for some time to describe plastic deformation (i.e., having a yield stress and strain hardening) in metals. Similarly, the equations of plasticity cannot describe failure in fatigue. Does that mean that metals do not deform plastically? Does that mean that we need a new paradigm and must throw out constitutive descriptions of plasticity? I think the answer to both questions is "no" since, although we need additional physics to describe fatigue, there are many situations in which a plastic description of metals is totally appropriate. Similarly, I'd argue that there are many situations in which the description of rocks as viscous fluids is totally appropriate.

So, at the very least, I suggest the authors weaken their comments to note that rocks can reasonably be treated as viscous fluids in many situations.

As a final note, although the formation of pore sheets may require additional physics in our constitutive models, in many cases, these additional physics can be described as an evolving viscosity. A relevant example is the work of Holtzman, King, and Kohlstedt (EPSL, 2012), who present a framework for describing the evolution of the viscosity of partially molten rocks as planar features of high porosity are formed.

To summarize my primary comments above, I think the wavelet analysis is an important contribution, and the observation of anisotropic pore distributions will motivate much future study. However, I feel the interpretations of these data are overstated since 1) whether or not porosity increases is unclear, 2) it seems the effects on permeability cannot be assessed from these data, and 3) a departure from viscous behavior associated with the pore evolution is not demonstrated. Based on these comments, I suggest the manuscript would require major revision to be accepted.

–Minor comments:

Line 20: It's not clear to me that the term "frictional embrittlement" is appropriate here. Is the implication that friction leads to brittle deformation, in the manner that hydrogen embrittlement means hydrogen doping leads to brittle behavior? I have been unable to find another use of this term in the literature.

Line 23: This does not seem like an appropriate reference here. The book by Kocks, Argon, and Ashby is certainly a classic work, but it is primarily focused on the small-scale aspects of dislocation motion. Their treatise is focused on the mechanisms that lead to plastic deformation (as in deformation with a yield stress), and I don't think they refer to geological processes, viscous behavior, or creep.

Line 25: It seems strange to refer to these rocks as viscous when the primary argument of the paper is that we can't think of rocks as viscous.

Line 35: The authors state here that the evidence is unambiguous for the role of creep cavities. This phrasing seems a bit strong to me considering my main comments above.

Line 54: At this point in the text, I was looking for a description of the sample preparation and data collection procedures. I eventually found a lot of this in the Appendix, but I think this part of the main text should point the reader to the relevant sections of the Appendix.

Figure 1: It would be very useful to know what the starting material looks like. In other words, does the porosity evolve simply as a function of increasing the confining pressure and temperature?

Figure 1: It would also be useful to state in the caption how the images were collected (e.g., BSE images in SEM).

Line 55: I'm curious how the authors distinguish between a pre-existing second phase and a new precipitate. And on a related note, if some pores are filled with precipitates, then are they still considered pores in this analysis? Based on the text here and in the appendix, it is unclear to me if, and if so how, secondary phases are removed from the porosity density maps.

Figure 2: How come a similar analysis for the low-strain material isn't shown? That would be a useful comparison.

Figure 2: It took me a while to figure out that eta is a sort of measure of the total power for the whole map. It would be helpful to clarify this in the main text and point the reader to section A7.

Figure 2: Both panels b and c have three peaks labeled, but only two are discussed in the text.

Lines 73 and 74: The values given for the locations of the peaks seem too precise to me considering the broadness of the peaks. I suggest reducing the precision by at least one significant digit. Or present some measurement of error for these numbers.

Figure 3: Panel g seemed out of place to me here as it doesn't really have to do with the wavelet visualization. Isn't it better suited to Figure 1?

Figure 3: It took me a while to figure out how exactly the visualizations were made. Much of this information is in the appendix, but more detail can be given here. The caption can simply state that these images represent the convolution of the density map and the wavelet, or something along those lines.

Line 83: The authors state here a key thrust of the paper, that the development of anisotropic porosity is not accounted for in our conceptual models. But the question that arises to me as I read this line is...do we need to account for it? I think that need is what remains to be demonstrated. Is the permeability measurably affected? Is the mechanical behavior affected? The mechanical data were collected and published for these experiments. Is there in signature in those data of the porosity evolution?

Line 87: If an attempt is made in a future revision to quantitatively relate the observed porosity to permeability, then this seems like a good spot for a back-of-the-envelope calculation.

Line 113: The text states that "...the hydro-mechanical anisotropy presented here would...". However, hydromechanical anisotropy is not presented here. Microstructural anisotropy is presented, and the hydro-mechanical anisotropy is only inferred.

Line 138: Again, how does the algorithm deal with secondary phases? Are they initially marked as pores?

Line 138: Here and elsewhere "S" is used instead of "A" to indicate items in the appendix.

Line 161: There is a typo of some sort in this sentence.

Line 185: There is some explanation here, but the choice of L still seems pretty subjective. Isn't it simpler to just retry the analysis with different values of L and see how that affects the results?

Line 215: I'm a little confused about the definition of "mean power spectrum". Is this the mean of the power spectrum? The mean of several power spectra?

Section A6 and Figure A4: I think this is a useful analysis of the edge effects. Some aspects of this analysis could be highlighted in the main text. For instance, in Figure 3, the caption could note that the white space around the visualizations indicates the region subject to edge effects, the size of which is dependent on the wavelength. This also begs the question, for the maximum wavelengths investigated ($\sim$600 microns), what proportion of the input image is actually useable?

---

## Author Comment (AC1) · 6 Nov 2020

Dear Editor,

Please find below our replies to the referees' comments and a revised manuscript that incorporates changes based on the feedback received.

The main change to the manuscript is the inclusion of an expanded discussion about the relationship between the porous anisotropy we observe in our experiments and the mechanics of mylonites. Referee 1 found that there was a logical gap in our previous text when it came to how and when the pore sheets would have a mechanical impact.

[Figure]

This was echoed by several questions and comments from Referee 2. We agreed with the referees' perspectives and have provided a larger discussion. We kindly thank the referees for their constructive comments and hope that they find our expanded discussion is a more complete treatment.

Minor changes include a new figure (formerly fig. 3g) and some clarifications over what we are suggesting to be reappraised. In the initial submission we think that we worded some sentences poorly and gave the wrong impression about what our opinions were in relation to the constitutive models of irreversible physical deformation in rocks. Additionally, some of the consequences of including creep cavities in the general shear zone model that were formerly included within the text (and a few more examples) have now been given their own section. We have also relabelled the minerals in the new figure after some new analysis conducted since the initial submission (we have now included this in the appendix).

We have attempted to remain in keeping with the nature of a short communication and we hope that you feel that the revised manuscript achieves this while addressing the referees' concerns.

Best wishes, James Gilgannon

Please also note the supplement to this comment:
https://se.copernicus.org/preprints/se-2020-137/se-2020-137-AC1-supplement.pdf

**Supplement:**

Dear Editor,

Please find below our replies to the referees' comments and a revised manuscript that incorporates changes based on the feedback received.

The main change to the manuscript is the inclusion of an expanded discussion about the relationship between the porous anisotropy we observe in our experiments and the mechanics of mylonites. Referee 1 found that there was a logical gap in our previous text when it came to how and when the pore sheets would have a mechanical impact. This was echoed by several questions and comments from Referee 2. We agreed with the referees' perspectives and have provided a larger discussion. We kindly thank the referees for their constructive comments and hope that they find our expanded discussion is a more complete treatment.

Minor changes include a new figure (formerly fig. 3g) and some clarifications over what we are suggesting to be reappraised. In the initial submission we think that we worded some sentences poorly and gave the wrong impression about what our opinions were in relation to the constitutive models of irreversible physical deformation in rocks. Additionally, some of the consequences of including creep cavities in the general shear zone model that were formerly included within the text (and a few more examples) have now been given their own section. We have also relabelled the minerals in the new figure after some new analysis conducted since the initial submission (we have now included this in the appendix).

We have attempted to remain in keeping with the nature of a short communication and we hope that you feel that the revised manuscript achieves this while addressing the referees' concerns.

Best wishes,

James Gilgannon

###########################################################################

In the following, referee comments are in grey and responses are in black.

Please find a copy of the manuscript with changes marked below the replies.

###########################################################################

Response to the comments of Alberto Ceccato (Referee 1):

Lines 13–14: please add some references supporting this statement.

Some example references have now been added.

Lines 32–34: I partially disagree with these two sentences. It is true that much of the past advances on the creep cavitation subject results from experimental works and micro–scale analysis (as the cited references report); but it is also true that many other authors have evaluated the extension and occurrence of creep cavitation and related phenomena in natural polymineralic shear zones deformed at geological conditions and have tried to qualitatively extrapolate their results from the thin section–scale to crustal– scale shear zones (e.g. Giuntoli et al., 2020 SciRep https://doi.org/10.1038/s41598- 020-66640-3; Preciguot et al., 2017 NatComm https://doi.org/10.1038/ncomms15736).

We have changed the text accordingly to reflect this and it now points the reader to a few other examples.

I understand that your samples are "natural" geo–materials, but you are presenting the results of experimental deformation at lab conditions, not geological conditions. I suggest the Authors to recast these two sentences taking into consideration this observation. I would focus on two main points that, in my opinion, are the strong points and the novelty of the paper: 1) the quantitative approach to evaluate the extension of creep cavitation processes; 2) the occurrence of creep cavitation in monomineralic shear zone.

This is an excellent point and we agree that the experiential conditions of any laboratory deformation places limits on the interpretation of the results obtained. However, in principle the model of Fusseis et al. (2009) is a Generalised Thermodynamical model that is driven by the coupling of specific dissipative length scales that leads to an entropic steady state and therefore can, to an extent, be tested outwith of 'natural' deformation conditions. To do this we use these experiments as windows on an entropic steady state with the caveat that the processes activate are comparable to those that would activate in nature.

So, while it was never quantified in the original experiments, the experiments of Barnhoorn et al. (2004) are taken to qualitatively meet this thermodynamic requirement of steady state: that is the system underwent adjustment to accommodate the imposed deformational work and attained a mechanical and microstructural steady state. As the dissipative processes observed in the microstructure are similar to those seen in nature (e.g. creep accomplished by the migration of dislocations, sub-grain rotation

recrystallisation) it can be assumed that, despite the higher rate of work being done, the experimental boundary conditions keep the rock in a window that is still representative of how the rock sample could dissipate deformational energy at mid-crustal conditions when the system was closed. In this way, we are testing aspects of the Generalised Thermodynamic hypothesis of Fusseis et al. (2009) and do not find the difference in experimental conditions to nature as problematic as it may seem at a first pass. For this reason we have retained our original formulation as we wished to focus on the physical predictions and physical models.

Line 40: please, specify what you have revisited of the set of classical experiments (the microstructures of the deformed samples).

We have changed the text accordingly.

Line 55: please, specify if you have considered these grain–filled pores as "porosity" during the following density quantification or not. I guess you have considered it as "porosity", but this is not clear neither from the main text, nor from the Method section (or perhaps I missed it).

In the quantification only open pore space is mapped. There will be some pores that are partially filled with a precipitate but it is only the open space that is segmented and counts towards the porosity. We have clarified the text in the segmentation and wavelet analysis sections of the appendix accordingly.

Line 89: I would replace "microstructure of . . ." with "the spatial arrangement of syn–kinematic pores". In my opinion, the definition of the effective microstructure would imply the analysis and characterization of pore typology, connection, morphology, etc. . .

We have changed the text accordingly.

Line 94: Even though I agree with the Authors about the importance of the spontaneous development of a systematic pore pattern, this statement is rather speculative and not fully supported by the data presented in the manuscript. "Bulk material properties" is rather generic and vague. What are these bulk material properties and initial heterogeneities? Grain size? Mono–polymineralic composition of the sample? (What about the experimental boundary conditions? Shear strain rate? confining pressure? Sample/Room humidity and fluid availability?) I understand that this could lead to further discussion that goes beyond the scope of this short communication, but I would like the Authors to be a little bit more specific on the term "bulk material properties" if they want to keep the sentence. On the other hand, if the Author want to work around this

comment: highlight the lack of any initial heterogeneity and state that, at present, the genetic causes of this microstructure need further investigations. . .

Thank you for pointing this out, we agree that it is not clear as to what we were trying to express. We have changed the text to be less ambiguous and reflect the referee's suggestion. However, we would like to refrain from expounding on exactly which material property could be responsible. Our data does not allow us to say more than the observation of a porous anisotropy seems to be relevant to the sample scale (ie millimetres). With regards to what we consider material properties and heterogeneities we have provided some examples in the text but consider the wide spread definitions of what these are generically in the geological textbooks (e.g. Twiss and Moores, 2007; Fossen, 2010) to be something that we do not need to define in the manuscript. For the purposes of this reply we will say that we consider a material property to be an intrinsic property of the bulk material that is used in a macroscopic description of deformation. Furthermore, it is intensive and does not depend on the amount of the material. Scalar descriptions of microstructural features, like an average grain-size, we consider to be structural properties that describe the state of the material (cf. Kocks et al., 1975). As structural properties generally vary through the rock they can be one source of the kinds of initial heterogeneities that we refer to.

Twiss, R.J., & Moores, E.M. (2007). Structural geology. New York, NY, W.H. Freeman.

Fossen, H. (2010). Structural geology. Cambridge, Cambridge University Press.

Kocks, U., Argon, A., and Ashby, M.: Thermodynamics and kinetics of slip, vol. 19 of Progress in Materials Science, Pergamon Press Ltd, 1975.

Lines 104–106: These two sentences are rather speculative and vague. Unfortunately, it seems to me that a logical connection between the occurrence of strain invariant mechanical parameters and the possible evolution of pore sheets into creep fractures is missing. In my opinion, this paragraph suffers from two main problems: (1) there are no direct logical links between the observation of strain invariant mechanical parameters and the evolution of pore sheets into creep fractures, i.e. the maintenance of strain invariant mechanical parameters is not a sufficient conditions to prove (or just speculate) that also your pore sheets will evolve into creep fractures. (2) Even though the experiments of Dimanov et al. and Rybacki et al. have been performed under similar conditions to those reported here, there are some fundamental differences that might undermine your inference: a) different lithologies; b) occurrence of creep fractures at relatively low strains if compared to shear strains obtained in Barnhoorn et al.; c) lack of any evidence of creep

fractures in the set of experiments of Barnhoorn et al. (2004). Rather than a speculation on the possible evolution of these pore sheets under "variable" natural conditions, I would suggest the Authors to discuss and compare in a more detailed manner their experimental results and inferences with those presented by Dimanov et al (2007) and Rybacki et al. (2008). Then, a speculative extrapolation to the natural conditions might be attempted. Indeed, there might be some microstructural similarities between the cited references and the data reported in the manuscript that might better support your speculation about the "possible evolution of pore sheets into creep fractures". In addition, "natural conditions are more varied" is a rather vague and generic statement. What are these variable natural conditions? Transient shear strain rates? Wet vs. dry conditions? Please specify if you want to keep the sentence.

We agree with the referee's perspective (which is mirrored in the comments of Referee 2) and have substantially changed the discussion about pore sheets and the mechanics of mylonites.

We now provide a discussion on how the effect of boundary conditions, a more comprehensive note of the differences and similarities of those comparable experiments and others (Dimanov et al. (2007), Rybacki et al. (2008), Rybacki et al. (2010) and Delle Piane et al. (2008)). While still done in a speculative way, we now discuss some natural conditions more precisely. In general we have kept the same argument but have added some of the logic that had brought us to our initially more brief paragraph about the mechanics of shear zones and how pore sheets could impact on this.

Rybacki, E., Wirth, R. and Dresen, G., 2010. Superplasticity and ductile fracture of synthetic feldspar deformed to large strain. Journal of Geophysical Research: Solid Earth, 115(B8).

Delle Piane, C., Burlini, L., Kunze, K., Brack, P. and Burg, J.P., 2008. Rheology of dolomite: Large strain torsion experiments and natural examples. Journal of Structural Geology, 30(6), pp.767-776.

Technical Corrections:
Figure 1: please, enhance the brightness/contrast of the BSE images OR enlarge the images (this might help the reader to instantaneously capture the porosity distribution even if the image is very small).

We have enhanced the image to better show the contrast between features. We have also made a new figure (formerly fig. 3g) that shows the detail of a pore sheet. We would rather not change the scale of figure 1's panels as part of what we wish to emphasise is that the porosity is a feature that is relevant at the sample scale and is not just a local phenomenon.

Figures in the supplement: There is a discrepancy between the figure call–out in the text (Figure S*) and the Figure captions (Figure A*). Please change one or the other to be consistent throughout the text.

Thank you for pointing this out, we have changed the text to reflect the figure captions.

Line 245: There's probably a reference typo [(18) ???].

It was, thank you, we have changed it to the correct reference.

#########################################################################

Response to the comments of Lars Hansen (Referee 2):

–The nature of the cavities:
My question in this context is whether or not there is a pore fluid, and if so, what is its composition? One item of concern is whether the pores are filled with CO2. That is, could there be some decarbonation of the calcite during initial pressurization and heating that generates porosity? Caristan, Harpin, and Evans (1981) presented evidence of calcite decarbonization during hot-pressing of calcite aggregates in a gas- medium apparatus, and it seems worth discussing whether or not something similar has occurred here. This question could be answered in some part by comparing the starting material (not shown) to the low-strain material (gamma = 0.4).

We have amended part of the text in discussion section 3.1 to include some inferences about the possible composition of the fluid, which we agree likely has $CO_2$ as a component. It would, of course, be more satisfying to know the exact composition of the fluid that filled the pores, but this is beyond the scope of our study and we feel that the impact and importance of the new results presented is not undermined by a lack of knowledge of the fluid composition.

Regarding the straight forward calcite decarbonation reaction, which we do not address in the text directly, we think that it is likely not occurring as it is not thermodynamical favourable for the deformation conditions (e.g. Ivanov and Deutsch, 2002; Shatskiy et al., 2018). However it is very likely that the minor amounts of dolomite present in the Carrara

marble samples will allow a release of $CO_2$ through the reaction Dol <-> Cc + Per + $CO_2$ (cf. Delle Piane et al., 2008).

Ivanov, B.A. and Deutsch, A., 2002. The phase diagram of CaCO3 in relation to shock compression and decomposition. Physics of the Earth and Planetary Interiors, 129(1-2), pp.131-143.

Shatskiy, A., Podborodnikov, I.V., Arefiev, A.V., Minin, D.A., Chanyshev, A.D. and Litasov, K.D., 2018. Revision of the CaCO3–MgCO3 phase diagram at 3 and 6 GPa. American Mineralogist: Journal of Earth and Planetary Materials, 103(3), pp.441-452.

Delle Piane, C., Burlini, L., Kunze, K., Brack, P. and Burg, J.P., 2008. Rheology of dolomite: Large strain torsion experiments and natural examples. Journal of Structural Geology, 30(6), pp.767-776.

–The evolution of porosity and its measurement:
Considering that some porosity may be present at the onset of deformation, the fundamental question is whether or not the porosity increases with deformation. A primary conclusion of the previous work (Gilgannon et al., GRL, 2020), which analyzes one of the same samples as in this paper (PO422, gamma = 5), is that porosity increases due to pore nucleation associated with recrystallization. Indeed, the previous work presents excellent observations of pores associated with new grain boundaries, and a total porosity of ~1%, but as far as I can tell, it is never demonstrated that deformation increases the total porosity.

The present work presents an excellent opportunity to make that demonstration since samples deformed to a variety of strains are analyzed. However, it is still not clear to me that there is a change in the total porosity. The wavelet analysis definitely demonstrates a change in the spatial distribution of pores, but as far as I understand, it does not present an increase in the porosity.

My conclusion here stems from the details of the measurement of porosity density, which is the fundamental measurement presented in the paper. As I understand the measurement procedure described in the appendix, only the centroids of identified pores are used in the construction of the porosity density maps. Thus, "porosity density" is apparently the number of pores per unit area, and this quantity may be better referred to as the "pore density". I admit I had some confusion about how "standardized density" is defined, why it is unitless, and why it can have negative values. Some clarification here would be valuable in a revised version, but for now it seems to me to represent a

normalized version of the "pore density" and not the porosity (pore volume per total volume).

We thank the referee for the feedback and we agree that pore density is a more correct way to refer to the quantity and have changed the text and figures accordingly.

To the point of the standardised, and unitless, density: this occurs through the implementation of equation A7 where the value of each pixel is divided by the standard deviation of the image which has the same units as the pixel values of the image. We do this for the purposes of standardising the histograms of each pore density raster. This is because the wavelet analysis benefits from analysis on a field that has a mean value of zero (see reply fig. 1 below) and is an advised step for the correct use of the wavelet convolution.

[Figure]

Reply fig. 1: The histograms for both the raw and standardised values of the various kernel density density maps.

We must doubly thank the referee: because of his question we realised that we had mistakenly not retained the floating point information on the rasters and our calculations had been made on greyscale values rather than the correct density units. This did not affect our results, as it was simply a transformation of the values in the density field, but we recalculated each step of the wavelet analysis to be sure and our figures now present this correct data: there was no change in the results. We now use arrows in the visualisation of the pore density figures to show that the colour bar is clipped and other higher and lower values exist.

To the point of whether or not porosity increases: in the initial submission of the text we only referred to the spatial extent of the porosity (which does increase) and avoided discussing if the absolute porosity changed at all. The absolute porosity does in fact increase and based on the referee's desire to know this information we have chosen to now include it (now reported in the new table 1 of the manuscript).

We were reluctant to include this in the initial submission because image segmentation has many hard-to-account-for uncertainties and because there may have been grain boundaries included within our particular choice of workflow. After some consideration we are only willing to include these values because we note that the change in porosity between gamma 5 and 10.6 is on an order of magnitude. As the seed based segmentation (randomwalker) algorithm used the same diffusion gradient and dark seed value for each image it is likely that this change of an order of magnitude is real. Additionally as the pore shape filter used is the same between data sets and the pore shapes are not expected to change significantly between strains we expect that we are comparing comparable pore populations and hence syn-kinematic porosity values.

Unfortunately, the "pore density" does not seem as valuable as porosity in regards to interpreting the hydromechanical effects of the observed microstructural evolution. A key inference in the paper is that permeability is enhanced (and becomes anisotropic) during deformation. The relationship between permeability and porosity is not trivial, but if the porosity were known, a back of the envelope calculation could be made to assess the potential changes in permeability expected. However, I don't think changes in permeability can be assessed qualitatively or quantitatively when using only the number of pores. Furthermore, from the data presented here, it seems entirely possible that the porosity could be constant with increasing deformation, even though the number and distribution of discrete pores evolves.

As our paper seeks to test aspects the dynamic granular fluid pump model (Fusseis et al., 2009; Regnenauer-Lieb et al., 2009) and hence discuss the results in this context, it is not possible to calculate the permeability expected to be produced by the pump simply from porosity values. This is because the permeability is time dependent and a result of both mechanical and chemical dissipation. This makes the kind of back-of-the-envelope calculation for permeability not possible. In this sense one has to solve a full system of coupled non-linear equations to make a prediction of what the permeability is in the model of the dynamic granular fluid pump model. The strength of our results is that they observationally validate some of the predictions of the model of Fusseis et al. (2009) and provide the first unambiguous evidence in favour of the well cited but little tested dynamic

granular fluid pump model. We agree that it would be of interest for future work to attempt such a calculation and model the consequences of the increase in porosity and its extent.

Regenauer-Lieb, K., Yuen, D.A. and Fusseis, F., 2009. Landslides, ice quakes, earthquakes: a thermodynamic approach to surface instabilities. In Mechanics, Structure and Evolution of Fault Zones (pp. 1885-1908). Birkhäuser Basel.

As a final note on this topic, there is a clear link to previous work on the segregation of melt during deformation of partially molten rocks (for a review, see Kohlstedt and Holtzman, AREPS, 2009). I'm surprised to not see any of that work referenced in the present manuscript. Those studies demonstrate that the (melt-filled) porosity can dynamically arrange into planar features not dissimilar to the "periodic pore sheets" described here. However, I'll emphasize that the average porosity is static in those experiments. Locally it may increase or decrease, some new pores are created, some old ones are destroyed, but the total porosity does not change. It remains to be demonstrated that a similar phenomenon is not occurring in the calcite samples analyzed here and that the overall porosity is actually increasing.

We agree that there is a strong connection to the topic but we have refrained from a discussion in this direction for two reasons: (1) other work has tackled this exact comparison (Spiess et al., 2012) and (2) our experiments are considered to reflect the behaviour of single phase aggregates. There is much to discuss when considering only the implications for sub-solidus deformation in single phase aggregates and any more discussion about how the system would behave with melts would not let us keep to the short communication format we have submitted. We hope that other future contributions by the community will pick up on the work already started by Spiess et al., (2012) in considering the role of creep cavities and melt segregation.

Spiess, R., Dibona, R., Rybacki, E., Wirth, R. and Dresen, G., 2012. Depressurized cavities within high-strain shear zones: their role in the segregation and flow of SiO2-rich melt in feldspar-dominated rocks. Journal of Petrology, 53(9), pp.1767-1776.

–Challenging the concept of rocks as viscous fluid:

The authors take the discussion beyond the development of a periodic array of pores to comment on whether or not rocks can be treated as viscous fluids. The paper is framed around the common treatment of rocks as viscous fluids at long timescales and suggests that that framework is flawed and a new paradigm involving creep cavitation is necessary. Emphasis is given to the transition from viscous to brittle behavior.

I suggest that these statements are overstating the case and detract from what is otherwise a useful paper about applying wavelet analysis to periodic pore distributions. My primary concern here is that the authors do not provide any evidence that the mechanical properties of the rocks investigated are modified by the formation of an anisotropic pore distribution. The mechanical data have already been published in previous work, so if these pore sheets are significant to the rheological behavior, then the authors can demonstrate that with the mechanical data. Furthermore, if these rocks can't be described as viscous fluids because of the porosity evolution, then the authors could again demonstrate that with the mechanical data.

In addition, I'll note that viscosity is a phenomenological description, and when discussed in the context of crystalline materials, is generally only taken to apply at macroscopic scales. For example, we often discuss rocks as fluids with non-Newtonian viscosities for which the viscosity is controlled by the dynamics of dislocations at the lattice scale. There has been much work in the materials sciences over the past several decades demonstrating that, at scales well below the grain size, dislocation propagation occurs in discrete bursts (key terms are "dislocation avalanches" and "jerky flow"). This behavior is clearly not viscous, but when averaged over many crystals of many orientations, the mechanical behavior can still be described as viscous.

Key to the authors' argument is that creep cavitation can lead to brittle behavior, which "can never be predicted by flow laws commonly used to model viscous deformation." An analogous case can be seen in the fatigue of metals. Constitutive equations have existed for some time to describe plastic deformation (i.e., having a yield stress and strain hardening) in metals. Similarly, the equations of plasticity cannot describe failure in fatigue. Does that mean that metals do not deform plastically? Does that mean that we need a new paradigm and must throw out constitutive descriptions of plasticity? I think the answer to both questions is "no" since, although we need additional physics to describe fatigue, there are many situations in which a plastic description of metals is totally appropriate. Similarly, I'd argue that there are many situations in which the description of rocks as viscous fluids is totally appropriate.

So, at the very least, I suggest the authors weaken their comments to note that rocks can reasonably be treated as viscous fluids in many situations.

As a final note, although the formation of pore sheets may require additional physics in our constitutive models, in many cases, these additional physics can be described as an evolving viscosity. A relevant example is the work of Holtzman, King, and Kohlstedt

(EPSL, 2012), who present a framework for describing the evolution of the viscosity of partially molten rocks as planar features of high porosity are formed.

We think that we have worded some sentences poorly in the original submission and have given the wrong impression to the referee: our argument was not intended to claim that viscosity as a constitutive model is wrong. We have reworded text where we can to make this clearer. Our discussion that pertains to pore sheets and mechanics is significantly longer now and hopefully clearer. We have tried to fill in the logical gaps noted by Referee 1 and 2 in the original section of the text. Largely, our argument points to the same messages but we hope that the longer discussion helps the reader follow the argument more easily now.

The fact that a mylonite deforms in a fashion that can be described with a viscous model is something we agree with. Moreover we have no problem with any constitutive model. We wished to argue the case for the Generalised Thermodynamic model behind the work of Fusseis et al. (2009). We can see how the very short discussion in our initial submission coupled with some imprecise wording lead the referee towards his point.

In the Generalised Thermodynamic model behind the work of Fusseis et al. (2009) a mechanical rate equation is essential but it is not sufficient to fully describe mechanical dissipation. That is to say that we wished to open a discussion in the community about a perspective that does not limit a mylonite to solely slow aseismic creep. The strength of our work is that it shows there is some validity to at least one of the predictions of the dynamic granular fluid pump model (Fusseis et al., 2009). We wished to argue that the fact we found what is otherwise unexpected in our current perceptive of shear zones warrants discussion. This is especially true when there is another perspective, that of the dynamic granular fluid pump model, that predicts the observation should be there. Moreover, it is even more pertinent when a range of experiments show similar results.

Our contribution hopes to showcase the fact that quantitative testing validates parts of a paradigm that has consequences far beyond an esoteric porosity. Additionally, we fully agree with the concept of scales of deformation and the need for homogenisation of processes when considering their dissipative effects: this is explicitly part of the framework that makes up the Generalised Thermodynamic model we are arguing favourably for (cf. Regenauer-Lieb et al., 2013; Regenauer-Lieb et al., 2014; Veveakis and Regenauer-Lieb, 2014). We hope that our chief aim, that of arguing for the Generalised Thermodynamic paradigm concerning creep cavities, is clearer now in the text and the expanded discussion.

Regenauer-Lieb, K., Veveakis, M., Poulet, T., Wellmann, F., Karrech, A., Liu, J., Hauser, J., Schrank, C., Gaede, O. and Trefry, M., 2013. Multiscale coupling and multiphysics approaches in earth sciences: Theory. Journal of Coupled Systems and Multiscale Dynamics, 1(1), pp.49-73.

Regenauer-Lieb, K., Karrech, A., Chua, H.T., Poulet, T., Veveakis, M., Wellmann, F., Liu, J., Schrank, C., Gaede, O., Trefry, M.G. and Ord, A., 2014. Entropic bounds for multi-scale and multi-physics coupling in earth sciences. In Beyond the Second Law (pp. 323-335). Springer, Berlin, Heidelberg.

Veveakis, E. and Regenauer-Lieb, K., 2015. Review of extremum postulates. Current Opinion in Chemical Engineering, 7, pp.40-46.

–Minor comments:
Line 20: It's not clear to me that the term "frictional embrittlement" is appropriate here. Is the implication that friction leads to brittle deformation, in the manner that hydrogen embrittlement means hydrogen doping leads to brittle behavior? I have been unable to find another use of this term in the literature.

We have changed the text to remove this phrasing. It now reads:

It is this fracturing, which can have physical (e.g. Beall et al., 2019) or chemical (e.g. Alevizos et al., 2014) driving forces, that creates seismicity and mass transport pathways through the deep Earth (Sibson, 1994).

Line 23: This does not seem like an appropriate reference here. The book by Kocks, Argon, and Ashby is certainly a classic work, but it is primarily focused on the small- scale aspects of dislocation motion. Their treatise is focused on the mechanisms that lead to plastic deformation (as in deformation with a yield stress), and I don't think they refer to geological processes, viscous behavior, or creep.

We agree that, while it does treat homognised macroscopic deformation, the work of Kocks et al. (1975) is not the most geologically relevant citation. Therefore, we have changed the reference here to two more geologically relevant references:

Hobbs, B. and Ord, A.: Structural Geology: The Mechanics of Deforming Metamorphic Rocks, Elsevier, Netherlands, 2015.

Poirier, J.-P.: Creep of Crystals: High-Temperature Deformation Processes in Metals, Ceramics and Minerals, Cambridge Earth Science Series, Cambridge University Press, https://doi.org/10.1017/CBO9780511564451, 1985.

Line 25: It seems strange to refer to these rocks as viscous when the primary argument of the paper is that we can't think of rocks as viscous.

As discussed above, we apologise for the confusion but our primary argument was not to we cannot think of rocks as viscous. The text now reads:

While much of this paradigm remains to be tested, the notion that mylonites generate self-sustaining and dynamic pathways for mass transport is radical and consequential for the interpretation of how deep shear zones behave during deformation.

Line 35: The authors state here that the evidence is unambiguous for the role of creep cavities. This phrasing seems a bit strong to me considering my main comments above.

We have changed this part of the introduction and it now reads as:

In this contribution we provide unambiguous experimental evidence in a natural starting material that supports, and extends, the paradigm concerning the role of creep cavities in shear zones. We present quantitative results showing that creep cavities are a spatially significant feature of viscous deformation, being generated in periodic sheets throughout the samples. Our analyses are intentionally made over large areas of the experimentally deformed samples in order to contextualise and understand the role of creep cavities at a scale more comparable to those where macroscopic material descriptions are unusually made. We argue that our results warrant a reappraisal of the community's perception of how viscous deformation proceeds with time in rocks and suggest that the general model for viscous shear zones should be updated to include creep cavitation. A key consequence of this would be that the energetics of the defroming system become the keystone of our perspective rather than the mechanics.

Line 54: At this point in the text, I was looking for a description of the sample preparation and data collection procedures. I eventually found a lot of this in the Appendix, but I think this part of the main text should point the reader to the relevant sections of the Appendix.

We have added direction for the reader to find the methods now in section 2 of the manuscript.

Figure 1: It would be very useful to know what the starting material looks like. In other words, does the porosity evolve simply as a function of increasing the confining pressure and temperature?

Unfortunately we did not have access to the starting material. We consider the reference of a sample that has not experienced dynamic recrystallisation as adequate for the argument we wished to make.

Figure 1: It would also be useful to state in the caption how the images were collected (e.g., BSE images in SEM).

We have altered the figure caption to include this information.

Line 55: I'm curious how the authors distinguish between a pre-existing second phase and a new precipitate. And on a related note, if some pores are filled with precipitates, then are they still considered pores in this analysis? Based on the text here and in the appendix, it is unclear to me if, and if so how, secondary phases are removed from the porosity density maps.

We have changed the text in the methods section to reflect that we only segment open porosity.

Regarding the nature of new vs old second phases, we consider the fact that precipitates exist at the triple junctions of newly recrystallised calcite grains as a strong argument. Additionally, the precipitates often possess complex shapes that reflect the pore geometry is another strong argument. Please refer to the supplementary material of Gilgannon et al. (2020) (GRL) for a more complete argumentation of this point.

Figure 2: How come a similar analysis for the low-strain material isn't shown? That would be a useful comparison.

As we were interested in analysing the pore sheets, of which there were none before dynamic recrystallisation, the analysis does not include the low strain sample.

Figure 2: It took me a while to figure out that eta is a sort of measure of the total power for the whole map. It would be helpful to clarify this in the main text and point the reader to section A7.

We have altered the figure caption to include this information.

Figure 2: Both panels b and c have three peaks labeled, but only two are discussed in the text.

We have added text to the figure caption to explain why we do not discuss it. It reads:

We note that two local extremes are not discussed in the text. This is because one was at the sensible limit of the analysis ($\theta$ = -12°, $\lambda$ = 530 μm) defined in appendix section A6 and the other did not correlate with any microstructural features ($\theta$ = -84.00°, $\lambda$ = 173 μm).

Lines 73 and 74: The values given for the locations of the peaks seem too precise to me considering the broadness of the peaks. I suggest reducing the precision by at least one significant digit. Or present some measurement of error for these numbers.

We have now reduced the precision in the text to 2 significant figures.

Figure 3: Panel g seemed out of place to me here as it doesn't really have to do with the wavelet visualization. Isn't it better suited to Figure 1?

We have now made panel g its own figure.

Figure 3: It took me a while to figure out how exactly the visualizations were made. Much of this information is in the appendix, but more detail can be given here. The caption can simply state that these images represent the convolution of the density map and the wavelet, or something along those lines.

The figure caption has been changed accordingly

Line 83: The authors state here a key thrust of the paper, that the development of anisotropic porosity is not accounted for in our conceptual models. But the question that arises to me as I read this line is...do we need to account for it? I think that need is what remains to be demonstrated. Is the permeability measurably affected? Is the mechanical behavior affected? The mechanical data were collected and published for these experiments. Is there in signature in those data of the porosity evolution?

We agree that this is worth discussing and we have now substantially expanded the section pertaining to the mechanics to better treat this point.

Line 87: If an attempt is made in a future revision to quantitatively relate the observed porosity to permeability, then this seems like a good spot for a back-of-the-envelope calculation.

As discussed above, a back of the envelope calculation is not possible for the dynamic permeability in the dynamic granular fluid pump model which we test.

Line 113: The text states that "...the hydro-mechanical anisotropy presented here would...". However, hydromechanical anisotropy is not presented here. Microstructural anisotropy is presented, and the hydro-mechanical anisotropy is only inferred.

We agree and have changed the text accordingly.

Line 138: Again, how does the algorithm deal with secondary phases? Are they initially marked as pores?

We only analyse open porosity and as this has a different grey scale seed value the segmentation does not include second phases. We have changed the text to make it clearer that we only segmented and analysed open pore space.

Line 138: Here and elsewhere "S" is used instead of "A" to indicate items in the appendix.

Thank you for highlighting this. We have changed all instances of this mislabelling.

Line 161: There is a typo of some sort in this sentence.

We have changed the sentence accordingly.

Line 185: There is some explanation here, but the choice of L still seems pretty subjective. Isn't it simpler to just retry the analysis with different values of L and see how that affects the results?

Yes, as discussed in the appendix, it is convention when using wavelet analysis to choose a wavelet form that resembles the feature you are interested in. One could run the analysis through many iterations of the convolution with different kernels but this is very computationally costly. Currently we compute 60 scales and 181 orientations per pixel in the image and this requires a lot of memory. Future work may implement an automatic step to move through different kernels but it was not needed for our current purposes.

Line 215: I'm a little confused about the definition of "mean power spectrum". Is this the mean of the power spectrum? The mean of several power spectra?

We adopt the language used in the text of Torrence and Compo (1998). As we understand it, the mean power spectrum ($P_k$) is meant to be the mean value for the distribution of noise that would occur at a given frequency of the null model we use. In our case we use a white noise model (in which noise has the same power across frequencies). So the simplifying assumption of Torrence and Compo (1998) is that one does not have to

consider the distribution of noise at a given frequency but only what its mean value would be ($P_k$).

Section A6 and Figure A4: I think this is a useful analysis of the edge effects. Some aspects of this analysis could be highlighted in the main text. For instance, in Figure 3, the caption could note that the white space around the visualizations indicates the region subject to edge effects, the size of which is dependent on the wavelength. This also begs the question, for the maximum wavelengths investigated (~600 microns), what proportion of the input image is actually useable?

We have changed the figure text accordingly to mention the area of edge effects.

Regarding the proportion of the input image that is actually usable at the maximum wavelength, it is 30% of the original image's area. This is stated in the text of section A6.

[revised manuscript text omitted]

---

## Referee Report (RR1)

**Comments to the Authors for the manuscript se-2020-137-manuscript-version2**

I reviewed a previous version of the manuscript, and I am glad to see that the Authors answered in an exhaustive and very detailed way to the concerns of both Reviewers. The Authors' modification to the previous version of the manuscript effectively improved both the readability and the scientific content and structure of the paper. The overall length of the paper is right at the word count limit for a Short Communication. I wish to see these data and discussions published as soon as possible. Unfortunately, I still have some minor comments which the Authors may want to consider. The manuscript already fulfil the high-quality standards of Solid Earth and it deserves to be published after some very minor corrections. I provide below some comments and suggestions which I hope might help the Authors in shortening the text and clarify even more its content.

General comments:

1) Title: as it is now, the title is rather inconsistent with the whole bulk of topics discussed in the manuscript. I am referring to "focus mass transport", which subject is rather limited in the Discussion section. This comment relates also to the General comment 3) about Section 3.1.
Perhaps a slight rewording of the title would make it effectively reflect the content of the whole manuscript (e.g.: "Experimental evidence that viscous shear zones generate periodic pore sheets: effects on fluid redistribution and mechanical behaviour"; something that include both topics. By the way, this is only a suggestion.)

2) Introduction and Discussions: there are two seminal papers, in my opinion, from Neil Mancktelow (2002, "How ductile are ductile shear zones?" *Geology* https://doi.org/10.1130/G22260.1; and 2008 *Lithos* https://doi.org/10.1016/j.lithos.2007.09.013) which must be discussed (or at least cited) in your manuscript. Both papers cover exactly the topics and paradigms you wish to discuss, and thus I am quite surprised in not seeing them even cited in your manuscript.

Briefly, Mancktelow (2002) shows that there is the necessity of a "pressure-sensitive plastic deformation" component during viscous deformation of ductile shear zones to explain melt–fluid flow within ductile shear zones from a continuum mechanics point-of-view. Indeed, including this point-of-view in the Introduction (Lines 20-32) would strengthen and support your claim for the necessity of a "reappraisal of the community's perception of how viscous deformation in rocks proceeds with time", alongside with natural and experimental results. It would also support your discussion in Sections 3.1 and 3.2.3 (Lines 225-227).

3) "How mylonites could focus mass transport". I would like to see a clear discussion and separation between what is observed and inferred from the experimental data and microstructures and what is then extrapolated to occur in natural shear zones. I'll explain myself. The presented experimental data and microstructures show that there is the formation of a systematic, periodic and anisotropic pore network which allows for the mass redistribution within your sample, rather than an effective mass transport. The deforming sample in conjunction with the deformation apparatus cell constitute a "closed" chemical system.

Given that the "system definition" is a matter of scale, if one considers the deforming sample and the confining medium as two separate entities, the ingress of Ar from the confining medium is a clear evidence for the occurrence of an effective mass transport between two media. However, this cannot be demonstrated with the presented data and I completely understand that demonstrating Ar mobility is far beyond the scope of the present manuscript.

By contrast, mass transport in natural shear zones implies either gain or loss of chemical components in an "open" chemical system, which commonly includes two media: the shear zone and the some other rock (host rock, subducted or nearby tectonic units, e.g. Selverstone et al., 1991 JMG; Barnes et al., 2004 JMG) which act as either source or sink of the transported mass.

Therefore, I would suggest the Authors to clearly state that in the experimental case the porosity allows for a mass redistribution within the sample, which sample can be probably treated as a closed system. Then, if this process is extrapolated and adapted to the natural "open system" shear zones, where the deforming dynamic-porosity-bearing medium communicates with another medium, it can effectively promote mass transport. This can be easily addressed by the Authors with some rewording of the paragraph.

4) In my opinion, it would be better to present first the comparison with other experiments and then discuss how the porous anisotropy may affect the mechanics of such experiments (swop the order between section 3.2.1 and 3.2.2). This would also allow to extend the discussion about the Generalised Thermodynamic model to the other experiments and thus discuss the differences between the experimental results. Indeed, when reading the section comparing experimental results one question comes up: what about the boundary conditions (constant force vs. constant velocity) in these different experiments? Does it relates to the mechanics? This is only a personal suggestion, but it would probably ease the reading of the manuscript and its logical structure.

Otherwise, the Authors need to consider the boundary conditions as one of the "difference and similarities" between experiments (Lines 183-197), given that they show the important role of boundary conditions on the mechanics and microstructure of experiments in the previous section (3.2.1).

5) Section 3.3: I really like the discussion about veining and fluid flux, which perfectly fits with the above-discussed "mass transfer" capability of shear zones related to creep cavitation and the dynamic fluid pump model. The reference to "recent experiments on calcite gouges" fits perfectly with the "earthquakes and tremors" topic discussed at the end of the paragraph and the discussion of experimental mechanics above. However, the discussions concerning dyking and frictional melting seem a bit out of place, they are still speculative (as stated by the Authors) and a bit disconnected to the rest of the paper in my

opinion (Delete Lines 230-233; 236-240). Therefore, I would suggest the Authors to limit the discussion about "speculative" topics, also in order to shorten the main text. A rapid rewording and swop of sentences within the paragraph will easily satisfy this suggestion, if the Authors agree on that.

Detailed comments:

Line 80: If you are not considering the mineral-filled pores in your quantification, then specify that these values are minimum estimates.

Line 165: Please specify what is the related boundary conditions in the Generalised Thermodynamic model (i.e. constant velocity = constant thermodynamic flux; constant force = ?).

Bologna, 04.12.2020

Alberto Ceccato

---

## Author Response (AR2)

Dear Editor,

Please find below our replies (marked in light blue) to the referees' comments (marked in grey) and a copy of the manuscript that highlights the incorporated changes based on the feedback of the referees.

Best wishes,

James Gilgannon

Referee 1: Alberto Ceccato

I reviewed a previous version of the manuscript, and I am glad to see that the Authors answered in an exhaustive and very detailed way to the concerns of both Reviewers. The Authors' modification to the previous version of the manuscript effectively improved both the readability and the scientific content and structure of the paper. The overall length of the paper is right at the word count limit for a Short Communication. I wish to see these data and discussions published as soon as possible. Unfortunately, I still have some minor comments which the Authors may want to consider. The manuscript already fulfil the high-quality standards of Solid Earth and it deserves to be published after some very minor corrections. I provide below some comments and suggestions which I hope might help the Authors in shortening the text and clarify even more its content.

General comments:
1) Title: as it is now, the title is rather inconsistent with the whole bulk of topics discussed in the manuscript. I am referring to "focus mass transport", which subject is rather limited in the Discussion section. This comment relates also to the General comment 3) about Section 3.1.
Perhaps a slight rewording of the title would make it effectively reflect the content of the whole manuscript (e.g.: "Experimental evidence that viscous shear zones generate periodic pore sheets: effects on fluid redistribution and mechanical behaviour"; something that include both topics. By the way, this is only a suggestion.)

We agree that changing the title slightly would better serve the manuscript. It is now "Experimental evidence that viscous shear zones generate periodic pore sheets".

2) Introduction and Discussions: there are two seminal papers, in my opinion, from Neil Mancktelow (2002, "How ductile are ductile shear zones?" Geology https://doi.org/10.1130/G22260.1; and 2008 Lithos https://doi.org/10.1016/j.lithos.2007.09.013) which must be discussed (or at least cited) in your manuscript. Both papers cover exactly the topics and paradigms you wish to discuss, and thus I am quite surprised in not seeing them even cited in your manuscript.

Briefly, Mancktelow (2002) shows that there is the necessity of a "pressure-sensitive plastic deformation" component during viscous deformation of ductile shear zones to explain melt–fluid flow within ductile shear zones from a continuum mechanics point-of-view.

Indeed, including this point-of-view in the Introduction (Lines 20-32) would strengthen and support your claim for the necessity of a "reappraisal of the community's perception of how viscous deformation in rocks proceeds with time", alongside with natural and experimental results. It would also support your discussion in Sections 3.1 and 3.2.3 (Lines 225-227).

We agree and have exchanged some emails with Neil Mancktelow discussing this. Our initial exclusion came from citation limits imposed by other journals that we submitted the manuscript to prior to SE. We have now included citation of Mancktelow (2006) (https://doi.org/10.1130/G22260.1) and another important early work on creep cavities by Mancktelow et al. (1997) (https://doi.org/10.1007/s004100050379).

3) "How mylonites could focus mass transport". I would like to see a clear discussion and separation between what is observed and inferred from the experimental data and microstructures and what is then extrapolated to occur in natural shear zones. I'll explain myself. The presented experimental data and microstructures show that there is the formation of a systematic, periodic and anisotropic pore network which allows for the mass redistribution within your sample, rather than an effective mass transport. The deforming sample in conjunction with the deformation apparatus cell constitute a "closed" chemical system.

Given that the "system definition" is a matter of scale, if one considers the deforming sample and the confining medium as two separate entities, the ingress of Ar from the confining medium is a clear evidence for the occurrence of an effective mass transport between two media. However, this cannot be demonstrated with the presented data and I completely understand that demonstrating Ar mobility is far beyond the scope of the present manuscript.

By contrast, mass transport in natural shear zones implies either gain or loss of chemical components in an "open" chemical system, which commonly includes two media: the shear zone and the some other rock (host rock, subducted or nearby tectonic units, e.g. Selverstone et al., 1991 JMG; Barnes et al., 2004 JMG) which act as either source or sink of the transported mass.

Therefore, I would suggest the Authors to clearly state that in the experimental case the porosity allows for a mass redistribution within the sample, which sample can be probably treated as a closed system. Then, if this process is extrapolated and adapted to the natural "open system" shear zones, where the deforming dynamic-porosity-bearing medium communicates with another medium, it can effectively promote mass transport. This can be easily addressed by the Authors with some rewording of the paragraph.

We can see why this would be useful to clarify and have amended the text.

*We have changed:*
*"While it is unclear what the exact composition of the fluid was, the presence of many newly precipitated minerals in pores, and across pore clusters, is evidence that mass was mobile in porous domains during the deformation."*

*to read on lines 114-117:*
*"While it is unclear what the exact composition of the fluid was, the presence of many newly precipitated minerals in pores, and across pore clusters, is evidence that mass was mobile and redistributed during the deformation. Thus if extrapolated to a natural deformation where the chemical system may be more open, mass transport through this porous network could lead to mass gain or loss in the mylonite rather than just redistribution."*

4) In my opinion, it would be better to present first the comparison with other experiments and then discuss how the porous anisotropy may affect the mechanics of such experiments (swop the order between section 3.2.1 and 3.2.2). This would also allow to extend the discussion about the Generalised Thermodynamic model to the other experiments and thus discuss the differences between the experimental results. Indeed, when reading the section comparing experimental results one question comes up: what about the boundary conditions (constant force vs. constant velocity) in these different experiments? Does it relates to the mechanics? This is only a personal suggestion, but it would probably ease the reading of the manuscript and its logical structure.

Otherwise, the Authors need to consider the boundary conditions as one of the "difference and similarities" between experiments (Lines 183-197), given that they show the important role of boundary conditions on the mechanics and microstructure of experiments in the previous section (3.2.1).

We favour the current order of the sections but thank the author for his suggestion. We have added a clarification that the experiments we compare were also run at constant twist rates, like our own.

5) Section 3.3: I really like the discussion about veining and fluid flux, which perfectly fits with the above-discussed "mass transfer" capability of shear zones related to creep cavitation and the dynamic fluid pump model. The reference to "recent experiments on calcite gouges" fits perfectly with the "earthquakes and tremors" topic discussed at the end of the paragraph and the discussion of experimental mechanics above. However, the discussions concerning dyking and frictional melting seem a bit out of place, they are still speculative (as stated by the Authors) and a bit disconnected to the rest of the paper in my opinion (Delete Lines 230-233; 236-240). Therefore, I would suggest the Authors to limit the discussion about "speculative" topics, also in order to shorten the main text. A rapid rewording and swop of sentences within the paragraph will easily satisfy this suggestion, if the Authors agree on that.

We thank the referee for their suggestion but we wish to retain the inclusion of dyking and frictional melt. Firstly, this is because we have now linked this more clearly to melt segregation and flow, in line with the comments of Referee 2. Secondly, we feel that in such a discussion section about further consequences both the topics fit well.

As referee 2 rightly pointed out to us, our results are likely important for partially molten systems and while we are reluctant to focus the paper around this point we wished to include some mention of it in the more speculative aspects of our discussion. Additionally, in the case of frictional melting, and more broadly earthquake nucleation, it is clearly an important revelation if a mylonite is much more granular. It also fits perfectly with the sentiments of the citation (Mancktelow (2006)) the referee suggested we include about the modelling of mylonites as pressure-dependent rocks. We have changed this section now to link this discussion to the work of Mancktelow (2006). On lines 257-260 it now reads as:

*"This is relevant for observations of deep seated frictional melting where the presence of a porous anisotropy in mylonites may facilitate changes to some kind of granular or frictional mechanical mode that is otherwise unexpected. This would compliment earlier work that suggested that the mechanical behaviour of mylonites may be more pressure dependent than generally assumed (Mancktelow, 2006) and the emergence of creep cavities with high strains would facilitate this."*

Detailed comments:
Line 80: If you are not considering the mineral-filled pores in your quantification, then specify that these values are minimum estimates.

We thank the reviewer for the point they raise but we do not wish to make this change as it is much more complicated than simply saying there is a minimum due to some pores being filled.

Firstly, if the porosity is dynamic, as the model we test aspects of suggests, then we do not know if what we measure is a minimum, maximum, minima, maxima or any other position in the distribution of measurable values that could be occurring as pores open and close. It is likely that there is a very complicated competition of chemical and mechanical rates for pore opening and closing that determine how much pore space is open at any one moment (cf. the supp for Fusseis et a. 2009). Additionally, we would claim that regardless of whether the porosity was time-dependent, we would also need to know the porosity values of all other possible slices through the rock to know where our measurement sits in terms of a minimum value. We may in fact be observing a plane that gives porosity values much closer to a maximum. Moreover, the model of Fusseis et al. (2009) states that the opening of creep cavities would be proportional (in some way that is not clearly defined) to the strain rate, and as there is a rather large strain rate gradient through the

samples, with the max strain rate occurring within the window we sample, then we would in fact be much closer to a maximum for the sample. The comparison between the porosity values of our samples in our study is only really valid because the strain rates applied during the constant twist rates are broadly similar between experiments. Thus we compare very loosely 'constant strain rate windows' that vary with strain. We have included these porosity values only after the first round of referee's comments lead us towards this. There is a limit to how useful these values are and we do not wish to make that harder to evaluate by guessing at if they are close to a minimum or maximum value.

Line 165: Please specify what is the related boundary conditions in the Generalised Thermodynamic model (i.e. constant velocity = constant thermodynamic flux; constant force = ?).

We have amended the text accordingly.

Referee 1: Lars Hansen

The revised manuscript is greatly improved, and I thank the authors for taking the time to consider my previous comments. The introduction and discussion are much clearer in the revised version, and I find the inclusion of the absolute porosity changes to be intriguing. There are a couple of primary points I'd like to raise related to the revised text and to the response to my initial comments. I still think the content of the article is timely and worth publishing. My comments relate to the fact that I'm having trouble making some logical connections among different parts of the paper, and therefore some clarification would be welcome in a revision.

I admit that I did find the expanded discussion on the thermodynamics of creep to be difficult to follow at times. I appreciate the emphasis on the boundary conditions, and the possibility for constant velocity boundary conditions to prevent localization. However, I'm not sure I understand why a macroscopic response related to the porosity evolution must be linked to localization. There should still be some effect on the mechanical response associated with a 1% increase in the pore fraction, even if there isn't localization. This effect is well documented in the literature on partially molten rocks (e.g., see part 1 and part 2 of Hirth and Kohlstedt, 1995). Because of the reduction in grain-grain contact area, the local stresses are higher, and therefore higher strain rates are generated for the same macroscopic stress. The details of this reduction in apparent viscosity clearly depend on the specific microstructure, but for the basalt-olivine system in dislocation creep, you'd expect a reduction in viscosity of about 10% (factor of 1.5 increase in strain rate at constant stress, or a factor of 1.1 decrease in stress at constant strain rate, for a stress exponent of 3.5). Some of the experiments in Barnhoorn et al. (2004) may have weakening of this amount at strains >5, but the majority seem pretty stable.

We would contest that it is not immediately clear if the explanation given for why a partial molten system had a certain response during an experiment would directly apply to a sub-solidus deformation. That being said we agree that it is very surprising that the emergence of the porosity did not have any overall mechanical effect (like the referee described for partially molten systems). Possibly one of the most interesting implied results of our investigation is the apparent reinforcement of the sentiment of Dimanov et al. (2007) about the decoupling of the microstructure and the mechanics. In this sense, it points to a need to better understand the tension, noted in material sciences (cf. Rudnicki and Rice (1975) (https://doi.org/10.1016/0022-5096(75)90001-0) and Rice (1976) (https://www.osti.gov/biblio/7343664)), between material instabilities (where localisation arises due to macroscopic properties and processes) and geometric instabilities driven by material heterogeneities (imperfections, cracks, clusters of pores or a domain of finer grain size) in rocks. We hope that our results will spark future work in this direction.

I find it particularly interesting that the porosity doesn't increase until after gamma = 5. Considering that you are only counting open pores (as documented in the revised manuscript), this must mean the sample has increase in volume by 1% between a gamma of 5 and a gamma of 10. I understand the hypothesis that porosity is generated by recrystallization, but there is still plenty of recrystallization (perhaps even more) at low strains. So it remains unclear to me why there is a porosity increase only after very large strains (and as noted above, in association with essentially no change in flow stress). I think it would be useful to include some explanation, even if speculative, for this phenomenon.

Thank you for drawing attention to this interesting point. We too agree that it is something worth understanding but we cannot speak any more than we have already in the manuscript to the trajectory of porosity change with strain.

Unfortunately, there are far too many aspects of the experiments that are unconstrained to speculate in any way that would be meaningful or helpful for the reader. The main reason for why the porosity increases in such jumps is simply down to the resolution of our sampling. We chose samples that represented stages of the deformation and transformation and as such we cannot speak to the intermediate stages of porosity change as we do not have the data. From the data at hand we do not know, microstructurally, when the onset of significant dynamic recrystallisation was and can only assume that it must be close to peak stress. This, combined with the fact that the other samples arrested below peak stress show no to less than a few percent of recrystallisation (see fig. 6b in Barnhoorn et al. (2004)), suggests that our low strain sample must have had little to no recrystallisation occur. We would claim that this makes the comparison of the pore populations from before and after peak stress, from a process perceptive, limited.

Moreover, we agree that there are many interesting questions about what happens to the porosity values during the strains intermediate to peak stress and gamma 5. We would suppose to find creep cavities emerging and inherited porosity becoming overprinted with the new syn-kinematic porosity, but whether or not this means that there would be a transiently high, lower or similar porosity as the two pore population transitioned we cannot say. In the case of relating the post peak stress samples (gamma 5 sample to the gamma 10.6), this 1% porosity increase is achieved during a roughly 20-25% change in recrystallisation (see fig. 6b in Barnhoorn et al. (2004)). This is therefore not such a large jump in porosity value with strain/transformation by recrystallisation. While we too would like to know the details of this intermediate trajectory, it is beyond the scope of our current work and hope that future work will establish how these changes occur.

Regarding the link to partially molten rocks, I understand the authors' point that to properly treat that connection is a major undertaking. And I understand that this comparison has been made already by Spiess et al. (2012). However, I still feel that something needs to be said in this manuscript, even if it is just a single sentence referring the reader to Spiess et al. I should also note that the authors' second reason for not including this topic (that these experiments represent single phase aggregates) does not sound quite right to me. If there is a fluid phase in the pores (as the authors suggest there is in the revised version), then this is a multiphase system. Much of the relevant literature on partially molten rocks focuses on a single solid phase and a single fluid phase.

We completely agree with the referee that our work seems to lead to the conclusion that some single phase rocks may be more akin to those in multiphase systems. This was one of the major personal conclusions of the first author when writing up their PhD thesis, from which this work comes. However, we would claim that in the eyes of the community the experiments of Barnhoorn et al. (2004) are classical examples of how high homologous temperature deformation proceeds in monominerialic rocks and we would prefer to discuss our results with this in mind. As we have chosen a shorter communication, and because our actual results are purely those relating to pores and their distribution, we would rather keep to our current discussion where we guide the reader towards what it may mean for creep cavities and fluid to be present, be it melt or other liquids. In light of this, and because we agree with the referee, we have tried to accommodate a more direct link to the consequences of partially molten rocks. We hope that this draws the reader closer to the link that the referee hoped to establish.

The second paragraph of section 3.3 (Lines 245-252) now reads:

"*In the case of dyke intrusion at high grade conditions, the work of Weinberg and Regenauer-Lieb (2010) infact already invoked creep cavities and their coalescence into creep fractures as the responsible mechanism for allowing dyking to occur. While our results do not show the development of creep*

*fractures, they forward the speculative argument of Weinberg and Regenauer-Lieb (2010) that creep cavities will occur during ductile shearing in rocks and could be interpreted to add weight to the discussion of Spiess et al. (2012) about the grain-scale role of creep cavities in promoting melt segregation and flow. When this is considered alongside the body other seminal experimental work on partially molten rocks (e.g. Kohlstedt and Holtzman, 2009), it becomes clear that tests need to be devised to distinguish between the different theories of melt segregation and migration (e.g. compaction length vs. sheets of creep cavities)."*

I also appreciate the expanded discussion on how failure associated with cavity formation is not easily predicted in existing experiments in rocks. My comment here is intended to point out that there is an extensive literature on failure by cavity formation in metals, and the time to failure is a primary concern for engineering applications. Correspondingly, there are a variety of efforts to establish constitutive equations that predict the time to failure, which is exactly what the authors seem to be advocating for. Some well cited examples include Ashby and Dyson (1984, https://doi.org/10.1016/B978-1-4832-8440-8.50017-X) and Kowalewski et al. (1994, https://doi.org/10.1243/03093247V294309). As with the discussion of partially molten rocks, it seems unfortunate to not, at least, acknowledge that significant effort has already been made on this front outside the geological literature and refer the reader to this vast resource.

We have now amended the text to include reference to time to failure models. We thank the referee for raising their concern because it drew attention to nuance of our point that had been hidden in implication. The first paragraph of section 3.2.3 (Lines 212-221) now reads:

*"Our results, and those of the four other experiments described above, suggest that mylonites of various compositions develop complicated microstructures that for an unknown set of critical conditions can facilitate a spontaneous mechanical change from flow to fracture. While there are many ways to incorporate history-dependence into flow laws that account for some microstructural change (cf. Renner and Evans, 2002; Barnhoorn et al., 2004; Evans, 2005) it does not seem that this kind of rate equation would capture the differences between ours and the four other experiments described. For example, a flow law that integrated strain (e.g. Hansen et al., 2012) would not be able to account for why some of the five experiments fractured at shear strains below 5 (e.g. Dimanov et al., 2007; Rybacki et al., 2008, 2010) with others flowing up to a shear strain of 50 with no fractures developing (Barnhoorn et al., 2004). This observation also seems to highlight a limitation to the use of empirical relationships that link strain or time to failure by creep fracture (e.g. Rybacki et al., 2008). Taken together, we suggest that this potential disconnection between the microstructure and mechanics of a mylonite places a need for the use of more complex physics to describe how a shear zone may deform with time."*

The observation that some experiments fail and others do not seems to suggest that a damage mechanics approach to time to failure may overlook the dissipative competition of processes or misses some aspect about spatial distribution of damage. Even in the case of more complex "two damage state variable" approaches, like that of Kowalewski et al. (1994), they would likely not be able to reconcile the differences between the four torsion experiments we cited in our work. This is largely because they are not considering the energetic competition of processes, only the mechanical coupling of two linked damage variables. In the case of our experiments one could say that grain size change was some kind of damage with time and the creep cavities another coupled damage process. If one applied the assumptions of Kowalewski et al. (1994) our experiments should have failed at some point due the accumulation of damage. As they did not it seems to suggest that the steady state attained in our experiments must result from some more complex coupling of physics.

We have now highlighted our position more directly in the text and cite the geological work of Rybacki et al. (2008) as it also fits a time to failure curve. We would rather not include the mention of damage here as it has a rather vast literature base and one of the purposes of our short communication was to engage with parts of the tectonics community that may not have a clear background in rock mechanics. A reader interested in creep cavities would quickly find the damage mechanics literature and we do not think excluding citation to the material sciences here will be an impediment to the curious.

Minor comments:

Line 29: Why capitalize "Generalized Thermodynamic"? This can't be the only generalized thermodynamic model.

We have capitalised this because there are several schools of non-equilibrium thermodynamic frameworks and one of them is referred to formally as Generalised Thermodynamics. We have added a citation now to draw any curious reader to a review article in which this is defined.

Lines 29 to 30: Is it really a paradigm if much remains to be tested? I guess it sounds more like a hypothesis to me.

We would disagree because a paradigm more completely captures both a distinct set of ideas and the surrounding philosophy in which those ideas are pursued and tested. Contained within the paradigm is the notion of which data or observation is good or bad. In this sense it is as much about the perspective as it is the detail of the hypothesis. For example the microstructures of the experiments we revisit were originally investigated with a particular paradigm in mind which included assumptions about pores not opening at high confining pressure. We do not use the word paradigm to be superfluous but to convey holistically the nature of the dynamic granular fluid pump model which is about more than an esoteric porosity.

Lines 54 to 55: I'm not sure I fully understand this statement. Indeed, the energetics are critical to consider, but they are, to some extent, already included in the mechanics. For example, the typical quantities to measure in tests of viscous materials are stress and strain rate. The product of these two quantities is the total energy dissipation (in Watts per unit volume). Therefore, the energetics of the system are generally being measured and considered in traditional treatments of these data. Perhaps some additional clarification is necessary here.

We agree that the product of the stress and strain rate tensors will give a mechanical dissipation term, and the point we are making is that this term (along with other potential dissipation terms) must be included in the energy conservation equation for the corresponding coupled feedbacks to be accounted for.

Answering your question succinctly is best achieved by referring to fig. 4 in Herwegh et al. (2014) (From transient to steady state deformation and grain size: A thermodynamic approach using elasto–visco–plastic numerical modeling - https://doi.org/10.1002/2013JB010701). Here the referee will see that the rheological flow law is a mechanical rate that feeds into the energetic balance of the system but does not itself define the stability of the system. In more complex formulations of what is expressed in Herwegh et al. (2014) many dissipative processes can compete at several dissipative length scales (cf. Hobbs et al. 2011 (https://doi.org/10.1016/j.jsg.2011.01.013)). That is to say that when one considers the energy equation as the governing equation then rate equations (like flow laws (several may act together to dissipate energy) or reaction rates) feed into how temperature changes with time. Another key difference is that one also considers the stored energy of the system when placing the energetics first. Which, in an isothermal case you described, would recast your W/t = (sigma/t * epsilon) + (sigma * epsilon/t) to become W/t = F/t + Phi/t, where F is the Helmholtz free energy and Phi entropy production. Here the dissipation through a flow law would be captured in Phi/t. By considering these variables one is considering far more than one can in a rheological equation and places mechanical dissipation as one among many factors that govern the stability of a reacting and deforming system.

This information is readily available in the work we cite and we would rather not add more information within the manuscript to make the case for the details of this Generalised Thermodynamic framework. We hope that the referee finds this agreeable and agrees with us that we have provided enough choice citations for a curious reader to dive further into the topic.

Lines 58 to 59: T should be italicized.

The text has been changed accordingly.

Line 75: I think a slight rewording is needed to clarify that it is the clusters (rather than the pores) that are systematically oriented.

We have changed the text to:

*"The pore density map of this experiment highlights that pores appear in clusters and that these clusters repeat across a large area with a systematic orientation (fig. 1d)."*

Line 120: This statement caught me off guard a bit. I suppose my instinct is that the evolution of the porosity and porosity distribution will depend large on the nature of the pore fluid and the pore-fluid pressure. I think the only way to say that these features of the porosity depend on bulk material properties like the elastic modulus is to demonstrate that the porosity is different for different materials, rather than the suggestion that the porosity reaches steady state, as is implied here.

We have amended the text with an explanation and citation to address the referee's concerns. The text now reads:

*"Curiously, our results also seem to suggest that porous domains develop within zones of stable orientation and, possibly, wavelength (approx. 15° from the shear zone boundary with a wavelength of 400 µm). This is consequential because it implies that the emergence of porous sheets is determined by some bulk material characteristic (for example, like the elastic moduli) and not by the positions of any initial heterogeneities hosted within the starting material (akin to grain-size variations). This speculation about the role a bulk material characteristic controlling the location of pore sheet formation is possibly supported by other experimental observations that the regular spacing of creep fractures from the coalescence of pore sheets changed with temperature (see fig. 16 in Dimanov et al. (2007))."*

We agree with the referee that our contribution's results do not let us discriminate between which bulk characteristic could play a role but we now draw on the observation of Dimanov et al. (2007) that creep fracture spacing changed with temperature. This bulk change in microstructure across millimetres with a state variable further suggests that it is likely some kind of material scale control that governs the location and orientation of the pore sheets. It would be interesting to see if future studies can show for more than the two temperatures that Dimanov et al. (2007) did, if the change in spacing is linear or non-linear with temperature as this might illuminate if it is the elastic properties or not that are driving these changes.

Line 129: "affected" → "affect"

The text has been changed accordingly

Line 131: Remove "concerning" or "of"

The text has been changed accordingly

Line 148: Is "flux" the right word here? What is there a flux of? This discussion is about the boundary conditions, but flux refers to some sort of transport. Is this the flux of energy into the system? Some clarification would be useful.

As we draw on an existing definition we have now included citation for the reader to pursue further if they so desire.

Line 195: "evolved"→ "evolve"

The text has been changed accordingly

Line 198: Again, shouldn't the pore-fluid pressure factor into this discussion?

This is a good point and we have now amended the text to reflect the referee's comment. The text now reads:

*"It is not clear what critical condition leads some to fracture and others to not: for example, is it the local pore density, differences in pore fluid pressure or the widths of the porous domains that controls if a fracture develops?"*

Line 200: Remove "a" prior to "fracture"

The text has been changed accordingly

Line 230: An "e.g." makes sense before the citation to Hobbs since there are many documented observations of frictional melting in the geological record.

The text has been changed accordingly

[revised manuscript text omitted]